# Interactions between climate change, urban infrastructure and mobility are driving dengue emergence in Vietnam

Rory Gibb [1,2,3,4] ✉, Felipe J. Colón-González [1,2,3,5], Phan Trong Lan[6], Phan Thi Huong[6], Vu Sinh Nam[7], Vu Trong Duoc[7], Do Thai Hung[8], Nguyễn Thanh Dong[8], Vien Chinh Chien[9], Ly Thi Thuy Trang[9], Do Kien Quoc [10], Tran Minh Hoa[11], Nguyen Hữu Tai[12], Tran Thi Hang[13], Gina Tsarouchi[14], Eleanor Ainscoe[14], Quillon Harpham [14], Barbara Hofmann[14], Darren Lumbroso[14], Oliver J. Brady [1,2,3,17] & Rachel Lowe [1,2,3,15,16,17]

Dengue is expanding globally, but how dengue emergence is shaped locally by interactions between climatic and socio-environmental factors is not well understood. Here, we investigate the drivers of dengue incidence and emergence in Vietnam, through analysing 23 years of district-level case data spanning a period of significant socioeconomic change (1998-2020). We show that urban infrastructure factors (sanitation, water supply, long-term urban growth) predict local spatial patterns of dengue incidence, while human mobility is a more influential driver in subtropical northern regions than the endemic south. Temperature is the dominant factor shaping dengue's distribution and dynamics, and using long-term reanalysis temperature data we show that warming since 1950 has expanded transmission risk throughout Vietnam, and most strongly in current dengue emergence hotspots (e.g., southern central regions, Ha Noi). In contrast, effects of hydrometeorology are complex, multi-scalar and dependent on local context: risk increases under either short-term precipitation excess or long-term drought, but improvements in water supply mitigate drought-associated risks except under extreme conditions. Our findings challenge the assumption that dengue is an urban disease, instead suggesting that incidence peaks in transitional landscapes with intermediate infrastructure provision, and provide evidence that interactions between recent climate change and mobility are contributing to dengue's expansion throughout Vietnam.

Socio-environmental and climatic changes are reshaping the dynamics and distributions of infectious diseases worldwide, with urgent consequences for public health[1–3]. In recent decades these impacts have been especially pronounced for *Aedes* mosquito-borne arboviral infections (e.g., dengue, chikungunya and Zika), whose vectors are specialised for life in the emerging urbanised landscapes of the 21st

century[4]. Dengue is an acute febrile illness caused by any one of four major dengue virus (DENV) serotypes and is principally transmitted by *Ae. aegypti*, a human-specialist that breeds using water-related features of built environments (e.g., water containers in homes, gutters, drains and sewerage systems)[5,6]. The burden of dengue is rapidly growing, with incidence doubling each decade since 1990[7], cases reported from

more than 125 countries[8], and extremely widespread outbreaks increasing in frequency[5]. The disease is also expanding geographically into more remote regions[9,10], and to higher latitudes[11,12] and altitudes[13] at the margins of its historical range, which has complicated the historical perception of dengue as mainly a disease of major tropical cities[14]. These emergence trends are broadly thought to be driven by increasing human mobility[15,16], expansion of anthropogenic and semi-urbanised landscapes[10] and changing climatic suitability[17]. However, comparatively little is known about how these factors interact to shape dengue transmission and emergence patterns at the local scales relevant to outbreak surveillance and response.

Although cities are important regional foci of sustained DENV transmission and viral diversity in endemic areas[18–20], transmission patterns more locally are dependent on built environment characteristics that influence vector populations (e.g., housing quality, drainage, heat islands), and human movements that drive viral dispersal between locations[21,22]. Dengue burden is therefore highly spatially heterogeneous[23] (for example between neighbourhoods[24] or between major metropoles and smaller cities[15]), yet it is unclear which socio-environmental features are most influential in driving these variations in risk. Dengue is commonly associated with urban habitats[25], which provide both high densities of *Aedes* breeding habitat and amenable microclimates[6,26]. Urban growth is consequently often cited as a key driver of dengue transmission, but its effect is probably context-dependent; for example, expansion of built environments in the short-term may create many temporary open mosquito breeding habitats during the construction phase, and informal settlements (where infrastructure and services provision lag behind growth) may be more likely to increase risk compared to longer-term planned urban development. The link between dengue risk and water supply and sanitation infrastructure remains poorly understood, but these may be important factors determining spatial heterogeneity in transmission. Access to the piped water network should reduce households' need to store water in containers, and in household-level studies piped water access is often (but not always) associated with lower dengue risk[27–29]. Improvements in sanitation systems might similarly reduce risk by reducing the density of water storage containers; however, if not well-maintained, drains and septic tanks can be productive mosquito breeding sites[30]. Alternatively, human mobility patterns might be the dominant spatial driver of dengue. Well-connected hubs in international transport networks (e.g., metropoles or regional capitals) experience high rates of long-range DENV strain importation, seeding transmission chains that spread among closely linked areas via local traffic (e.g., commuter flows)[16,20,31]. Higher mobility might therefore be particularly important to the maintenance of dengue transmission in areas where epidemic fade-outs are more likely[32], such as with lower population densities or seasonally transient climatic suitability.

Climate has strong impacts on biophysical suitability for vector populations and dengue transmission. Air and water temperature affect numerous biological processes in mosquitoes that regulate population dynamics and vector competence (e.g., growth, survival, reproductive rate, extrinsic incubation period), which combined predict a nonlinear relationship between temperature and transmission intensity[26]. Temperature variability can underpin dengue outbreak seasonality[31] and transmission season length[33], and future warming temperatures are projected to significantly expand dengue transmission suitability worldwide[34]. However, there remains little evidence for how warming to date may have shaped recent dengue distribution and expansion trends. Precipitation patterns drive the creation and flushing of vector breeding sites[35], but their relationship to dengue transmission may often be nonlinear, delayed, and determined by how seasonality and extremes interact with local socio-environmental factors. For example, in Brazil and Barbados dengue risk sharply increases several months after periods of drought[36], particularly in urban areas with unreliable water supply[37], suggesting a mediating role of water

storage behaviour in response to rainfall shortages. These recent studies imply that local dengue responses to climatic drivers might differ markedly between neighbouring areas with different socioeconomic characteristics[23,38]. Further understanding such cross-scale interactions might improve the predictability of spatial outbreak dynamics in response to large-scale hydrometeorological phenomena such as droughts.

In this study, we investigate these interacting effects of climatic and socio-environmental drivers on dengue incidence and emergence in Vietnam, by analysing 23 years (1998-2020) of monthly district-level (2nd administrative level) case surveillance data. Dengue is a major public health issue in Vietnam which has among the highest incidence rates in Southeast Asia[39], although with wide variation in transmission intensity across its broad latitudinal and altitudinal range[40]. The south has a tropical monsoon climate and experiences fairly stable, seasonal endemic dynamics[40,41]. In the subtropical north, winter temperatures are too cool to support transmission[42–44] so dengue occurs in sporadic outbreaks during warmer months (often seeded by DENV reintroductions from the south)[42]. In recent decades, Vietnam has undergone a major economic transformation from low-income towards middle-income, with rapid development of major and regional cities, sharply rising population mobility via road and air (from ~3 million to ~53 million air passengers carried between 2000-2019[45]), and expansion of access to hygienic water supply and sanitation infrastructure to much of the population[46]. During the same period, the country has also experienced warming temperatures and more frequent extreme weather events such as heatwaves and drought and is considered particularly vulnerable to health impacts of climate change[47]. Currently, there is still little empirical evidence for how interactions between such rapid socioeconomic and climatic changes may impact the distribution and burden of dengue, making Vietnam an ideal historical setting to ask this question.

We used Bayesian hierarchical models and block cross-validation experiments to infer relationships between socio-environmental and climatic covariates and dengue incidence (Table 1), and explore their effects on spatiotemporal patterns of disease. We aimed to answer two main questions. Firstly, what are the most influential spatial and temporal drivers of dengue incidence across Vietnam, and how might these have contributed to recent dengue trends? Secondly, does local urban infrastructure modify the effect of hydrometeorological dynamics on dengue incidence?

## Results

### Surveillance data show a recent expansion of dengue incidence across much of Vietnam

Vietnam is administratively divided into 58 provinces and 5 major urban municipalities including its two main economic centres: Ha Noi and Ho Chi Minh City (Fig. 1). Since 1999 the country has maintained a national dengue passive surveillance system, with monthly reported case counts recorded at district-level (administrative level-2) (Methods, Supp. Fig. 1). Dengue incidence typically peaks between June and November (Supp. Fig. 1), so our analyses defined transmission years as running from May to April. The dataset included 174,936 monthly case counts totalling 2,038,380 clinically diagnosed (i.e., suspected) dengue cases, collected via passive surveillance from 667 districts between May 1998 and April 2021 (Methods). The highest country-wide counts were in 2019 (294,707) and 2018 (170,600), and the lowest in 2014 (34,258) and 2002 (35,386). Surveillance data show regional differences in transmission settings, with the south experiencing endemic dynamics, and the north sporadic outbreaks mainly restricted to Ha Noi and the Red River Delta (Fig. 1a and Supp. Fig. 1). Large synchronous outbreaks occurred nationally in 1998, 2010, 2017 (mainly in the north) and 2019 (mainly central and south) (Supp. Fig. 1). We mapped directional trends in dengue incidence at district-level by estimating the slopes of annual log incidence using linear regression (Fig. 1b). This

**Table 1 | Climatic and socio-environmental covariates as hypothesised drivers of dengue incidence**

| Covariate | Type | Source | Rationale |
|---|---|---|---|
| Annual temperature ($T_{mean}$, $T_{min}$, $T_{mean}$ of the coolest month) | Climate (temperature) | ERA5-Land[49] | Geographical limits on dengue virus persistence and transmission by mosquitoes[34]. Warmer annual temperatures are expected to facilitate year-round transmission. |
| Monthly temperature ($T_{mean}$, $T_{min}$, $T_{max}$) | Climate (temperature) | ERA5-Land[49] | Impacts spatial and seasonal biophysical suitability for DENV transmission[26]. Relationship may be nonlinear and depend on time delay. Tested at lags of 0 to 6 months. |
| Precipitation (monthly mm) | Climate (hydrometeorology) | WFDE5 v2.1.[50] | Impacts seasonal creation and flushing of *Aedes* breeding sites. Relationship may be nonlinear and depend on time delay. Tested at lags of 0 to 6 months. |
| Standardised Precipitation Evapo-transpiration Index (SPEI) in 1-month, 6-month and 12-month windows | Climate (hydrometeorology) | Derived from WFDE5 using 'spei' package[52] | Measures deviations from historical average hydrometeorological conditions for reference period 1981–2020 (i.e., excess or deficit), from short- to long timescales, so may be more sensitive to local context than simple precipitation. Relationship may be nonlinear and depend on time delay[36]. Tested at lags of 0 to 6 months. |
| Built-up land (proportion cover) | Urbanisation | ESA-CCI land cover (annual) | More built-up land is expected to increase availability of highly suitable *Aedes* habitat. |
| Urban expansion rate (km²/year over 3-year and 10-year window) | Urbanisation | Landsat urban dynamics[48] (annual) | Short-term (i.e., construction phase) and rapid or informal longer-term expansion of built environment may increase availability of suitable *Aedes* habitat. |
| Hygienic toilet access (proportion of households with indoor/outdoor flush toilet) | Infrastructure | Vietnam census 2009 and 2019 (interpolated to annual values) | Improved sanitation systems may reduce density of standing water for vector breeding sites, and therefore reduce transmission. |
| Improved water access (proportion of households with piped or borehole-derived water) | Infrastructure | Vietnam census 2009 and 2019 (interpolated to annual values) | Higher access may reduce propensity to store water around homes, reducing vector breeding sites and thus transmission. |
| Population density (persons per km²) | Population | Gridded Population of the World 2000 and Vietnam census 2009 and 2019 (interpolated to annual values) | Higher population density is expected to lead to increasing contact rates and potential for long-term persistence of transmission chains, so may increase incidence. |
| Road traffic per inhabitant (km travelled per inhabitant per year) | Mobility | VGSO (annual, province-level) | Higher rates of within-province population movements are expected to increase local dengue spread. |
| Potential population fluxes (mean gravity and radiation flux) | Mobility | Gravity and radiation models (applied to annual population) | Mean of pairwise predicted population fluxes between focal district and all other districts in Vietnam, based on population size and distance. Model-based proxy for relative attractiveness of districts for population movement (e.g., commuting). Higher movement rates are expected to increase rates of influence dengue introduction and spread. |

The table lists covariates used in models, their broad class, data sources, and rationale for testing. A fuller description of covariate sources, original data resolution and processing are provided in Supp. Table 1, Methods and Supp. Text 1.
*ERA5-Land* ECMWF Reanalysis v5 over land, *WFDE5* bias-adjusted ERA5 reanalysis precipitation data with reference to GPCC and CPC station data, *VGSO* Vietnam General Statistics Office.

shows strong evidence ($p < 0.01$) of upward trends throughout the southern central regions (South Central Coast, Central Highlands; up to a 45% year-on-year increase in some districts), Red River Delta, and parts of the Southeast (Fig. 1b). Standardised WHO dengue diagnosis guidelines and lab-confirmation practices were applied nationally throughout the study period, with no obvious step change in the data in 2009 when the WHO definitions were changed (Methods), suggesting that these trends are unlikely to be driven by a specific change in surveillance or diagnostic practices. The trends may still in part be driven by reporting factors, such as increased clinical awareness or healthcare access, although the pronounced geographical pattern and visual indications of a shift towards endemic dynamics in southern central regions (Supp. Fig. 1) are strongly suggestive of true expansion.

**National and regional trends in urbanisation, infrastructure, mobility and climate**
We derived district-level covariates to represent key hypothesised drivers, from census sources, remote sensing data[48], human mobility models and climate reanalysis (ERA5-Land temperature and bias-corrected ERA5 precipitation[49,50]; Table 1, Methods, Supp. Text 1). These included annual population density, built-up land extent, short-term and long-term urban expansion rates, improved water access

(piped or borehole-derived), hygienic toilet access (indoor or outdoor flush), mobility metrics (per-capita road travel rates, mobility flux predicted from naïve gravity and radiation models), annual temperature metrics to represent thermal constraints on dengue persistence (mean and minimum in the same dengue year), and monthly means of air temperature ($T_{min}$, $T_{mean}$ and $T_{max}$), precipitation, and multi-scalar drought indicators (Standardised Precipitation Evapotranspiration Index, SPEI[51], in 1-, 6-, and 12-month time windows) at lags of 0 to 6 months. SPEI measures accumulated hydrological surplus or deficit relative to the long-term historical average for the same period of the year[52,53]. Its multi-scalar nature enables measurement of hydrometeorological dynamics at timescales ranging from transient (affecting surface water) to long-term (affecting reservoir and groundwater levels), and thus different potential causal influences on dengue transmission (Table 1). Covariates, their definitions and hypothesised relationships are summarised in Table 1, with data sources and processing described in Methods and Supp. Text 1.

The study period saw nationwide upward trends in urbanisation, mobility and infrastructure improvement, although with regional variation (Fig. 2 and Supp. Fig. 2). Urban extent and urban growth, population density, mobility, and improved water and sanitation access are generally highest in the regions containing Vietnam's largest

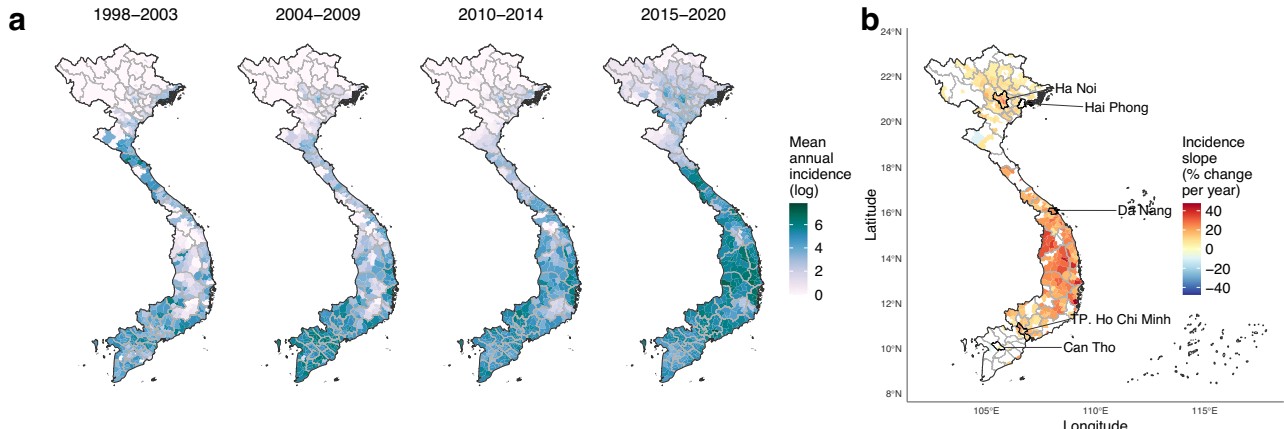

**Fig. 1 | Geographical distribution and trends in dengue incidence at district-level in Vietnam. a** Mean annual dengue incidence rates across all dengue years (May to April) within each 5–6 year time period between 1998 and 2020 (cases per 100,000 persons, log+1 transformed for visualisation purposes) for districts with dengue time series available (*n* = 667). Surveillance time series commenced between 1998 and 2001 depending on the region (see Methods). **b** Estimated slopes of annual dengue incidence rates between the earliest year of surveillance and 2020 (% change per year) are shown for districts with strong evidence (*p* < 0.01) of increasing (red) or decreasing (blue) trends, with Vietnam's 5 major urban municipalities labelled. Slopes were inferred using ordinary least squares regression without adjustment for multiple comparisons (as districts were not spatially independent from each other and the principal purpose was visualisation). The latitudinal gradient in seasonal dynamics is shown in Supp. Fig. 1.

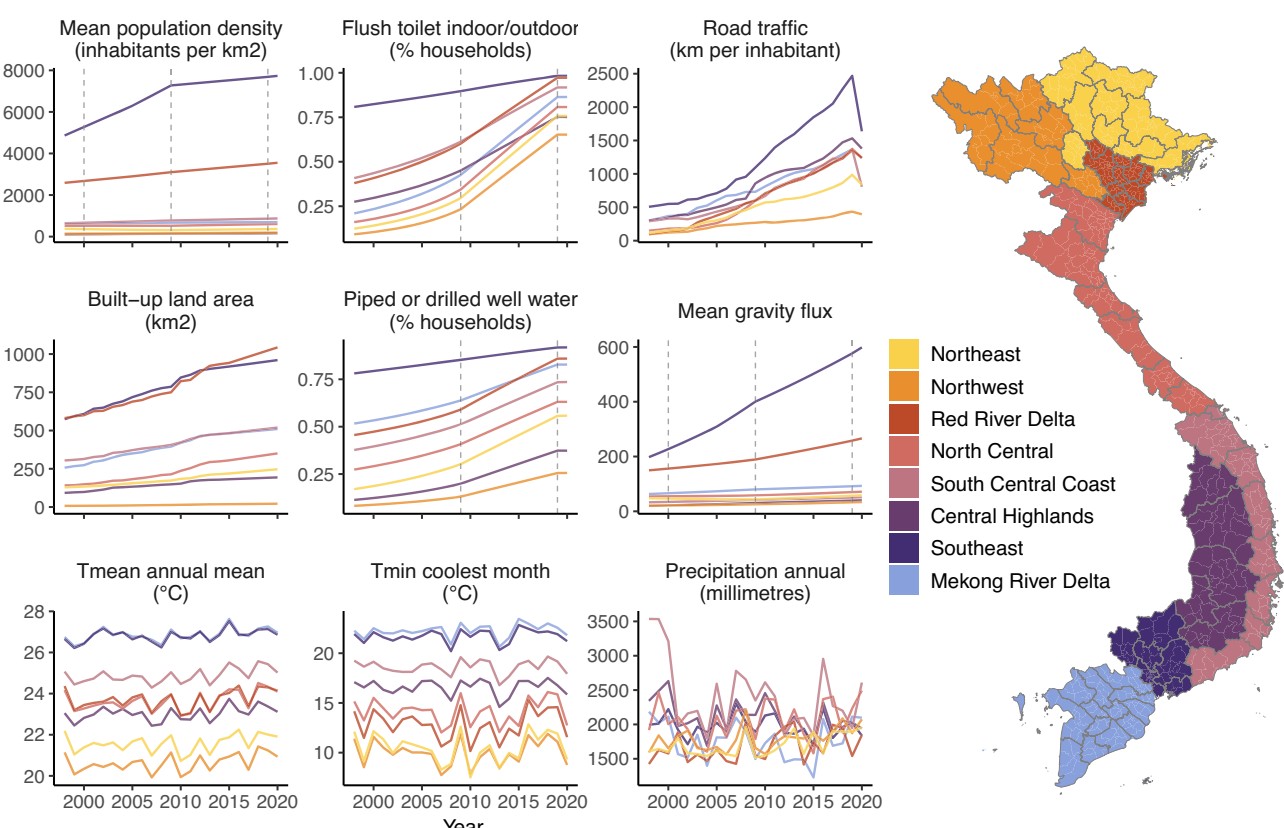

**Fig. 2 | Socio-environmental change and climatic variability in Vietnam from 1998 to 2020.** Sub-plots show annual socio-environmental and climatic covariate data (Table 1), aggregated from district- to region-level for the visualisation (region denoted by line and map fill colour; population density, gravity flux, temperature and precipitation are summarised as the mean across all districts). For census-based metrics (population, infrastructure, and gravity models) annual estimates were obtained via district-level interpolation or back/forward projection from observed years, which are shown as dotted lines (Methods, Supp. Text 1). For all other metrics, data were available for all years. Urbanisation, population density and mobility are highest in the subregions with the two largest municipalities: Ha Noi (Red River Delta) and Ho Chi Minh City (Southeast).

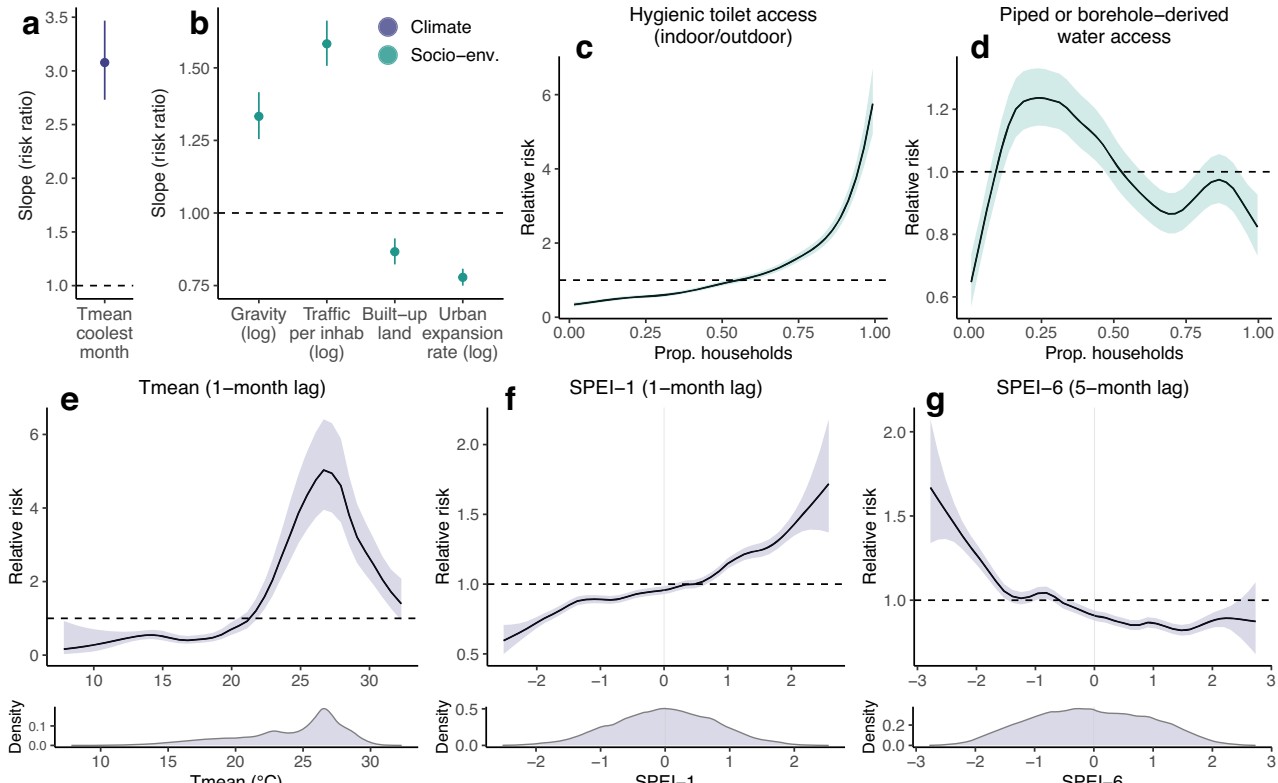

**Fig. 3 | Effects of socio-environmental and climatic drivers on district-level dengue incidence.** Sub-panels show posterior marginal linear fixed effects (**a**, **b**) and nonlinear effects (**c**–**g**) from the full-fitted model of district-level dengue incidence (*n* = 667 districts, 174,936 observations; Methods). Linear fixed effects are shown as risk ratios for scaled or log-transformed covariates (i.e., proportion change in risk for a 1 unit change in covariate), with points and error-bars showing posterior marginal mean and 95% credible interval. Nonlinear marginal effects (specified as second-order random walks; Methods) are shown on the relative risk scale, with lines and ribbons showing posterior mean and 95% credible interval. Point or ribbon colour denotes broad covariate class: either socio-environmental (green) or climatic (blue).

economic centres, Southeast (Ho Chi Minh City) and Red River Delta (Ha Noi). Per-capita road traffic rates (reported annually at province-level[54]) increased rapidly nationwide between 1998 and 2019—ranging from 4.5-fold growth in the Mekong River Delta and Northwest to 14-fold in the Red River Delta—and declined in 2020 reflecting COVID-19 associated movement restrictions. Census estimates also showed a nationwide expansion in the proportion of households reporting access to improved water supply (piped or borehole-derived water) and hygienic toilet facilities (indoor/outdoor flush toilet; this rose sharply from 2009 to 2019). Temperature becomes cooler and more seasonally variable along the south-to-north gradient, while precipitation is generally highest and most variable in the central regions (Fig. 2 and Supp. Figs. 2–4). Hydrometeorological extremes at short timescales (SPEI-1) are relatively variable among neighbouring districts (i.e., at small spatial scales), whereas at longer timescales (SPEI-6) they tend to be more spatially synchronised at the regional level (Supp. Fig. 3).

## Urban infrastructure, temperature and hydrometeorology are important spatial and seasonal drivers of dengue incidence

We fitted Bayesian spatiotemporal regression models to the surveillance dataset, with monthly case counts modelled using a negative binomial likelihood (Methods). Seasonality was represented with a province-specific temporally-correlated effect of calendar month ('*seasonal random effect*'). Unexplained spatiotemporal variation, for example, due to immunity and DENV serotype dynamics or changing surveillance sensitivity, was accounted for with dengue year-specific district-level spatially-structured and unstructured effects[55] ('*district-level random effects*'). A random effects-only ('*baseline*') model

captured declines in dengue relative risk (RR) and greater seasonal variability with increasing latitude (Supp. Fig. 5). We then tested whether socio-environmental covariates (specified as either linear, logarithmic or nonlinear terms) and monthly climate variables (at lags from 0 to 6 months) improved model adequacy metrics and reduced unexplained variation in district-level random effects, compared to the baseline (Methods, Supp. Fig. 6). There were greater improvements from including gravity rather than radiation model-based mobility flux; long-term urban expansion (in the preceding 10-year window) rather than short-term (3-year window); SPEI metrics rather than precipitation; and $T_{mean}$ of the coolest month rather than other annual temperature metrics (Supp. Figs. 6 and 7).

We developed a full multivariable model (Fig. 3, Methods) including fixed effects of $T_{mean}$ coolest month, built-up land cover, 10-year urban expansion rate (log), gravity flux (log) and road travel per inhabitant (log), and nonlinear effects of hygienic toilet access, improved water access and monthly $T_{mean}$ (1-month lag), SPEI-1 (1-month lag) and SPEI-6 (5-month lag). The full model substantially improved all information criteria (Supp. Table 2 and 3). Structured predictive experiments can provide insights into the generality of drivers, through identifying variables that improve predictive accuracy in unobserved locations and times[56,57]. To estimate the individual predictive influence of each covariate, we used 5-fold cross-validation to estimate model prediction error (out of sample mean absolute error, $MAE_{OOS}$) under 3 block holdout designs, in turn excluding one covariate at a time from the full model (Methods, Supp. Fig. 8). We defined a variable's 'predictive influence' as the change in $MAE_{OOS}$ when it is excluded (Fig. 4). We measured covariates' influence on predicting spatial heterogeneity in incidence using 'spatial' and

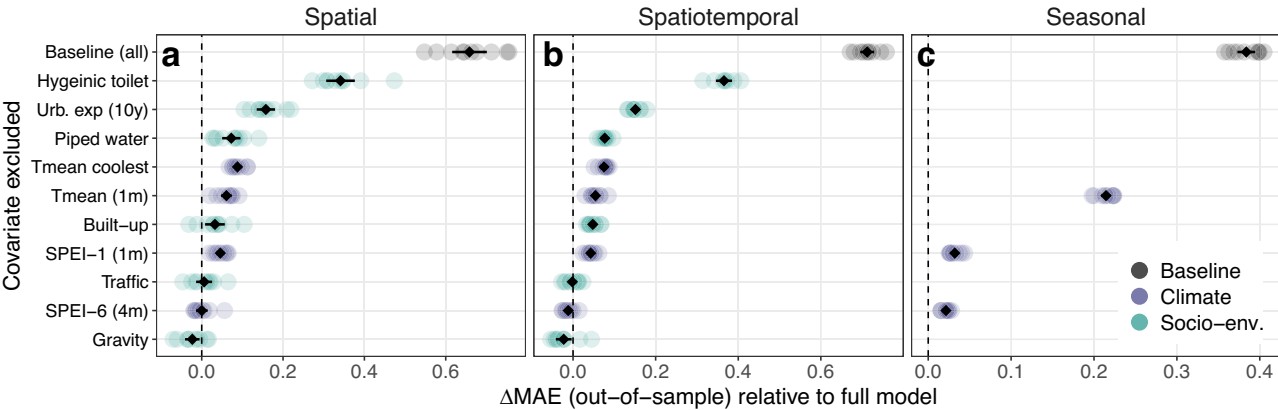

**Fig. 4 | Influence of individual socio-environmental and climatic factors on spatiotemporal and seasonal predictions of dengue incidence.** Influence of individual covariates on out-of-sample mean absolute error (MAE) was evaluated using 5-fold cross-validation under 3 block holdout designs: spatial (entire districts; panel **a**), spatiotemporal (district-year combinations; **b** and seasonal (quarterly blocks within each district; **c** (Methods, Supp. Fig. 8). Candidate models excluding one covariate at a time from the full model are shown on the y-axis, with the baseline (random effects-only) model for comparison. Individual points show change in MAE relative to the full model (dashed line), across 10 repeats account for variability due to random reallocation of cross-validation folds. Point colour denotes broad covariate class: socio-environmental (green), climatic (blue) or baseline model (grey). Black points and error-bars summarise the mean and 95% confidence interval across all 10 repeats. Values above zero indicate an increase in prediction error relative to the full model when a covariate is excluded (i.e., positive influence on prediction accuracy), and vice versa.

'spatiotemporal' block designs (5-fold blocked by district and district-year, respectively), and on predicting temporal dynamics using 'seasonal' block design (5-fold by district-quarter; monthly climate variables only) (Supp. Fig. 8). The full model significantly reduced prediction error compared to the baseline under all block designs (Fig. 4 and Supp. Fig. 9).

The most influential local spatial drivers of dengue risk related to infrastructure and urban expansion, followed by temperature and SPEI-1 (Fig. 4a, b). Increasing access to hygienic toilets had a positive marginal relationship with dengue risk (Fig. 3c) with the highest predictive influence (Fig. 4a, b). The effect of population access to improved water supply was nonlinear, with risk peaking at a low-to-intermediate level (around 25% of households) and declining thereafter (Fig. 3d). Urbanisation metrics had generally protective effects, with a strongly negative relationship between dengue risk and long-term urban expansion (in the preceding 10 years) with a high predictive influence (Figs. 3b and 4a, b), and a weaker negative effect of built-up land cover. Mobility metrics (per-capita road traffic rates and gravity flux) had positive relationships with dengue risk (Fig. 3b) but little overall predictive influence (Fig. 4). All inferred socio-environmental effects were robust to sensitivity analysis by census-defined level of urbanisation (Supp. Fig. 10).

Overall, there was strong evidence that temperature is a dominant factor shaping both the broad geographical distribution and temporal dynamics of dengue incidence across Vietnam. Annual $T_{mean}$ of the coolest month had a large positive effect on dengue risk and contributed significantly to spatial prediction (Figs. 3a and 4a, b), probably through impacting vector survival during the least thermally suitable period of the year. Notably, including this covariate alone reduced unexplained variation in the district-level random effects by 50%, providing strong evidence that thermal constraints on year-round DENV transmission by mosquitoes are a key determinant of the geographical gradient of dengue across Vietnam (Supp. Fig. 7). Monthly mean temperature ($T_{mean}$) had a nonlinear and delayed (1-month lag) effect, with relative risk increasing to a peak around 27 °C and declining sharply at higher temperatures, consistent with expectations based on dengue's thermal biology[26]. Monthly $T_{mean}$ contributed significantly to spatial prediction (Fig. 4a, b) and was the main predictor of temporal dynamics (Fig. 4c).

Hydrometeorological dynamics had delayed and nonlinear effects that depended on timescale: increases in relative risk were associated with transient excess wet conditions at short lead times (SPEI-1 1 month lag), and with long-term accumulated drought at longer lead times (SPEI-6 5-month lag) (Fig. 3f, g). SPEI-1 had a positive predictive influence on both spatial and temporal dengue dynamics (Fig. 4). In contrast, SPEI-6 did not substantially contribute to spatial prediction (Fig. 4a, b) despite improving temporal predictions (Fig. 4c). It is possible that the regional synchrony of long-term drought (Supp. Fig. 3) makes it less predictive of finer-scale spatial heterogeneity in dengue incidence, in the absence of information about local mediating socio-environmental features.

## The importance of human mobility is greater in northern Vietnam where dengue is emerging

The importance of drivers might vary between endemic contexts (i.e., where dengue persists year-round) and emerging settings, where sustained transmission is constrained by factors such as remoteness or transient climatic suitability. We examined this by fitting separate models for Vietnam's southern (Mekong River Delta, Southeast, South Central Coast and Central Highlands) and northern regions (North Central, Red River Delta, Northeast, Northwest), which broadly delineate areas of endemic and sporadic transmission (Fig. 1; Supp. Figs. 1 and 5). The inferred shape and directionality of socio-environmental effects were very similar between regions, albeit with generally larger fixed effects slope estimates in the north, reflecting the lower incidence of dengue compared to the national average (Supp. Fig. 11). The major notable difference is that mobility variables (per-capita road traffic rates and gravity flux) have relatively much larger positive effects on dengue incidence in northern Vietnam than in the endemic south (Supp. Fig. 11). The same regional differences are reflected in covariates' relative predictive influence under block cross-validation: the top-ranked spatial predictors in the north are mobility and temperature variables, compared to infrastructure, temperature and urbanisation in the south (Supp. Fig. 12).

## Climate change is reshaping the geography of dengue transmission across Vietnam

Ongoing climatic changes might be contributing to recent dengue emergence trends, particularly in the central and northern regions of

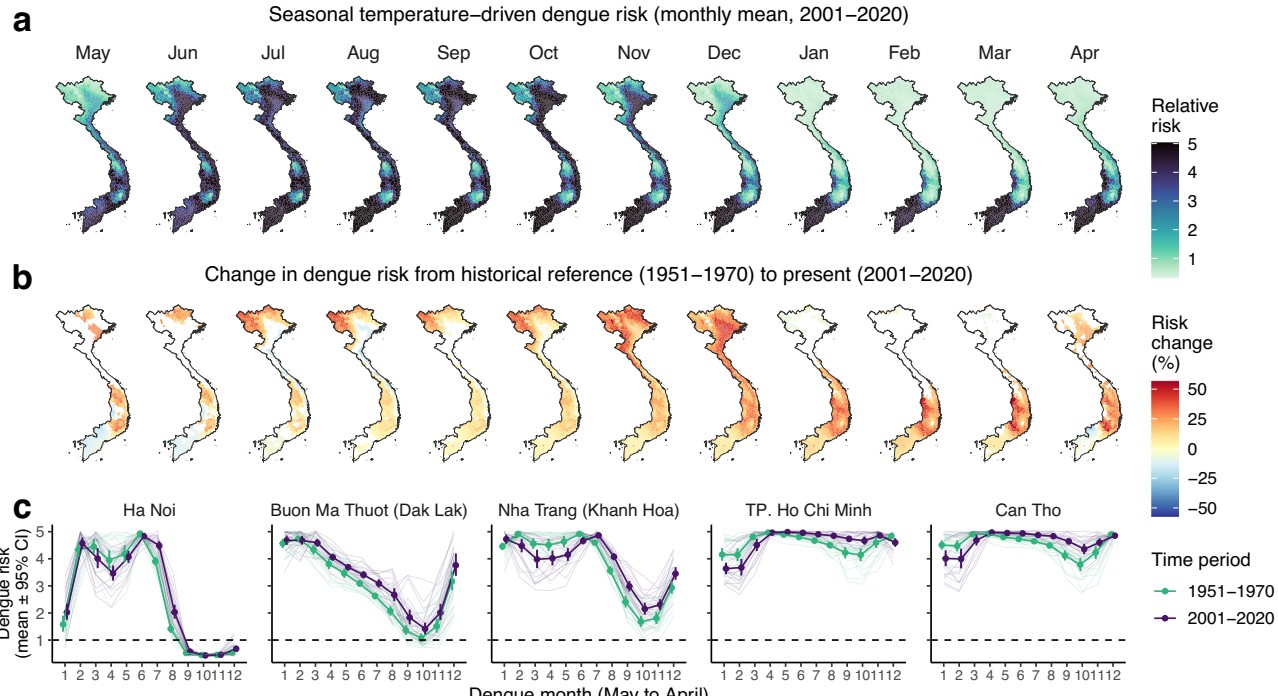

**Fig. 5 | Recent climate change has expanded and redistributed dengue transmission risk across Vietnam.** The full model was used to predict monthly marginal temperature-driven risk since 1950 using ERA5-Land reanalysis data (i.e., holding all other variables constant). Top maps **a** show present-day monthly 20-year means of dengue relative risk (2001–2020), with darker colours denoting increased risk. Bottom maps **b** show the monthly percentage difference in 20-year average dengue risk between historical reference (1951–1970) and present-day (red shading denotes increasing risk and blue decreasing risk). Only statistically significant differences ($p < 0.05$) are shown, with non-significant differences shaded white. Differences were estimated using categorical regression without adjustment for multiple comparisons (as districts were not spatially independent from each other and the principal purpose was visualisation). Graphs **c** show long-term changes in seasonal risk by dengue month (May to April) for 5 example cities with high dengue burden (Ha Noi in north; Buon Ma Thuot in the central highlands; Nha Trang on the south central coast; and Ho Chi Minh and Can Tho in the south). Fine lines show individual years, and points and error-bars show monthly 20-year mean and standard error ($n = 20$), with lines coloured by time period (green for reference period; purple for present-day). The supplementary material shows more district examples (Supp. Fig. 13) as well as changes in temperature patterns for these 5 localities (Supp. Fig. 14). Results were very similar when defining a later reference period (1971–1990; Supp. Fig. 15).

Vietnam (Fig. 1b). To investigate this, we tested for significant changes in monthly temperature-driven dengue risk between a historical reference period (1951–1970) and the present-day (2001–2020) using long-term ERA5-Land reanalysis data[58] (Methods). We used the inferred risk function for $T_{mean}$ (Fig. 3e) to predict monthly posterior marginal mean temperature-driven risk since 1950 (i.e., just the effect of temperature while holding all other variables constant), then used linear models to test for differences between reference and present-day periods, comparing 20-year averages to account for natural climate system variability (Methods, Fig. 5). Present-day projections of temperature-driven risk reproduce the gradient of observed transmission, with high risk year-round in the south, and seasonally transient risk in the north that declines during winter months (January to April; Fig. 5a).

Increasing temperatures since the 1951–1970 reference period have driven expansion and redistribution of predicted dengue risk across much of Vietnam (Fig. 5b, c; Supp. Figs. 13 and 14). Predicted risk increases are particularly pronounced in southern central regions, including during low-season months in the higher-altitude Central Highlands provinces (up to 56% increase), suggesting that climate change is expanding the suitable area for endemic transmission. Similarly, much of north Vietnam has experienced sharp rises in risk during summer months, including in more remote northern regions, and a lengthening of the transmission season in the Red River Delta including Ha Noi (Fig. 5b, c). Notably, these hotspots of increasing temperature-driven risk are geographically concordant with the steepest upward dengue trends during the 1998-2020 period (Fig. 1b) and

with visual indications of a transition from sporadic outbreaks towards endemic transmission cycles in southern central regions (Supp. Fig. 1). Overall, these results suggest that recent warming has reduced thermal constraints on dengue transmission in much of Vietnam and probably contributed to recent northward and altitudinal shifts. Notably, however, there is also evidence of seasonal and spatial redistribution of transmission, with rising temperatures above dengue's thermal optimum slightly reducing risk during the hottest months of the year in parts of the south (April-July) and north and coastal areas (July–August), compared to the historical reference period (Fig. 5b, c; Supp. Figs. 13 and 14). Far fewer climate observations are assimilated by ERA5 during earlier years (up to the late 1960s), which may impact the accuracy of reanalysis estimates[58]; as a sensitivity test we, therefore, repeated this analysis using a later reference period (1971–1990), which showed very similar overall results (Supp. Fig. 15).

## The effects of hydrometeorology on dengue incidence are multi-scalar and modified by local infrastructure

Theory and recent empirical evidence suggest that local socio-environmental context may be important in determining dengue's response to precipitation and drought patterns[35,37]. We investigated the multi-scalar effects of hydrometeorological dynamics on dengue incidence, focusing on southern Vietnam where transmission occurs year-round (Methods). We found evidence of delayed and timescale-dependent relationships between hydrometeorology and dengue risk: the increase in risk driven by transient wet conditions (SPEI-1) peaks at

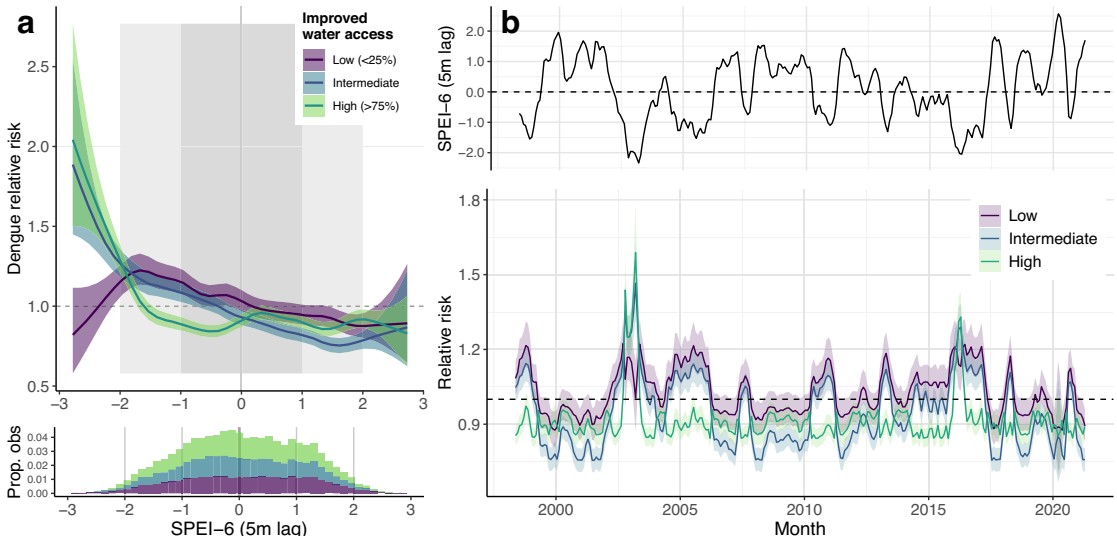

**Fig. 6 | Improved water supply modifies the effect of long-term drought (SPEI-6) on dengue incidence in southern Vietnam.** The fitted interaction between SPEI-6 (5-month lag) and piped or borehole-derived water access is shown in **a** (low = <25% of households; medium = 25–75%, high = >75%), with lines and shaded area showing posterior marginal mean and 95% credible interval. Histogram shows the distribution of observations across the 3 strata (bar height is cumulative across 3 strata). Visualisation of the marginal effect of SPEI-6 on relative risk is shown in **b**, for an example time series of SPEI-6 from the Mekong River Delta region (top row; Dong Thap province), under scenarios of low, intermediate and high improved water access (bottom row). Lines and ribbons show posterior marginal mean effect and 95% credible interval. Accounting for this interaction reduces predictive error under both spatiotemporal and seasonal cross-validation (Supp. Figs. 16 and 17).

a 1- to 2-month delay and declines sharply beyond 2 months, whereas risk associated with long-timescale drought (SPEI-6) emerges gradually over a longer delay period (from 4 to 6 months) (Supp. Fig. 16). This suggests that hydrometeorological phenomena at different timescales probably affect dengue risk via different causal pathways, one mainly biophysical (high rainfall leading to immediate proliferation of outdoor vector breeding sites) and the other behavioural (household water storage in response to perceived sustained shortages).

If the long-term drought effect is mainly mediated by water storage behaviour, we hypothesised that increasing population access to improved water supply (i.e., more reliable than rainwater[59]) would reduce the dengue risks associated with sustained drought but not with short-term excess (Methods). We also expected that, at this fine spatial scale, including an interaction with water supply would explain observed patterns better than an interaction with urbanisation (which was found to significantly modify drought effects in a recent coarser-scale study[37]). We tested this by stratifying the effects of either SPEI-1 (1-month lag) or SPEI-6 (5-month lag) by low (<25%), intermediate (25–75%) and high (>75%) levels of either improved water supply (proportion of households) or built-up land cover, within the full model retaining all other covariates (Methods). Consistent with our expectations, models including an interaction between SPEI-6 and water supply substantially improved model fit, whereas interactions with urban land and with SPEI-1 did not improve models (Supp. Figs. 17 and 18). The interaction model showed a complex, nonlinear relationship between sustained drought, improved water supply and dengue risk (Fig. 6a). Increasing access to improved water supply reduces the delayed dengue risk associated with near-normal to moderately dry conditions (SPEI-6 between 0 and −1.5), but sharply increases the risk under moderate to extreme drought conditions (SPEI-6 < −1.5). In contrast, where improved water supply is low, increasingly dry conditions are associated with linear increases in risk except during rare periods of severe drought, when risk slightly declines (SPEI-6 < −2, 1.5% of observations). Long-term wet conditions (SPEI-6 > 0) are protective across all strata (Fig. 6a). Notably, including this interaction substantially reduced prediction error under spatiotemporal and seasonal holdout designs compared to a non-interaction

model (Supp. Fig. 18), providing evidence that accounting for cross-scale climatic and socio-environmental interactions can help to predict spatial variation in dengue risk.

To visualise how interactions between drought and improved water supply could produce spatial heterogeneity in dengue dynamics, we projected monthly SPEI-6 associated relative risk for an example time series from the Mekong River Delta (2002–2020), under scenarios of low, intermediate and high improved water supply (Fig. 6b). Over two decades, SPEI-6 oscillates between periods of near-normal to moderately dry and wet, with rarer extremes (droughts in 2003–4 and 2016–17, excess in 2018–19; Fig. 6b, top panel). Under low-to-intermediate improved water supply (<75% of households), dengue relative risk is closely linked to these oscillations, significantly increasing during regular dry periods. In contrast, when improved water supply is high (>75%) risk is effectively dampened during these lower amplitude dry periods, instead only increasing sharply during sporadic periods of severe drought (Fig. 6b, bottom panel).

## Discussion

Despite recognition of the growing threat of dengue under global change[5], understanding of how key socio-environmental and climatic drivers shape both local patterns of transmission and broader emergence trends remains patchy. By analysing 23 years of dengue surveillance data across a gradient of transmission intensity in Vietnam, we found that urban infrastructure-related metrics (water supply, sanitation and long-term urban growth) are the most influential predictors of local heterogeneity in incidence (Figs. 3 and 4). Notably, temperature is a key driver of dengue's distribution and dynamics, and long-term reanalysis data indicates that recent climate change has already expanded temperature-driven dengue risk across Vietnam (Fig. 5). In contrast, effects of hydrometeorology depend on timescale and socioeconomic context, with drought effects mediated by access to improved water supply (Fig. 6). These socio-environmental findings complement existing household- and subregional-level evidence for dengue risk factors in Vietnam[29,40,41,60], with the benefit that the dataset's long-term nature, spatial granularity and national coverage allowed for inference across the full range of many hypothesised

drivers—from rural to urban, remote to highly connected, and tropical to cooler subtropical climates.

## Decomposing the roles of urbanisation and infrastructure as spatial drivers of dengue incidence

The coarser surveillance data commonly used in large-scale dengue analyses makes it difficult to disentangle the effects of local socio-environmental factors from closely correlated metrics such as population density. Our study at finer-scale avoids this issue, and provides evidence that water and sanitation infrastructure are more important spatial determinants of dengue risk than availability of urban habitat per se. Increasing hygienic toilet access was the strongest positive predictor, which was unexpected as sanitation improvements are thought to decrease household-level risk. It is possible that this metric may instead index "urban-like" water-related infrastructure that provide amenable *Aedes* breeding habitat, such as storm drains, septic and water storage tanks (although this relationship was robust in a model fitted only to data from rural districts; Supp. Fig. 10), and/or proxy for economic improvements and associated increases in healthcare access that could increase case detection. Alternatively, increasing flush toilet access could itself drive population-level risk: indoor flush toilets in Vietnam are conventionally linked to septic tanks with storm drain overflows[46], and outdoor latrines often contain stored water containers, both of which provide vector breeding sites[30]. Indeed, previous studies in Vietnam have identified outdoor latrine access[41] and proximity to sewage discharge sites[61] as significant household-level risk factors for DENV exposure; our results suggest these effects might be more general, and highlight the need for further research to understand the role of changing sanitation systems in dengue emergence. We also found evidence for protective effects of high coverage of improved water supply in both south and north Vietnam, even though our metric was coarse (including both piped and groundwater-derived sources) due to limitations of census data (Fig. 3 and Supp. Fig. 2). This is consistent with evidence from Vietnam and elsewhere[62,63] and is likely mediated by the ability and propensity to store water around homes when water supply is low or unreliable (see below).

Notably, the negative effects of long-term urban expansion and built-up land suggest that—after accounting for mobility and infrastructure—dengue incidence declines in increasingly urbanised landscapes. This appears counterintuitive given that urban growth is typically considered a key dengue driver (although systematic reviews have not shown clear empirical consensus for this relationship[6,25]). However, our satellite-based metrics are probably better indicators of formally planned urban developments than of more informal or peripheral settlement expansion, which can be harder to detect from space[64]. Such developments may generally have better provision of water, sanitation and vector control services; as such, these results are consistent with our infrastructure findings in suggesting that socio-environmental characteristics are the key determinants of heterogeneity in risk across large areas. Taken together, these results suggest that dengue incidence in Vietnam probably peaks in semi-urbanised or peri-urban areas—i.e., relatively well-connected localities with extensive landscape modification for essential sanitation, water and drainage, but lacking higher-quality infrastructure and services that could otherwise reduce vector densities. This conclusion is supported by an earlier cohort and modelling study from Vietnam, which suggested susceptibility to large dengue outbreaks is highest in areas with intermediate population densities and low piped water access[62].

## Interacting effects of the climatic and socio-environmental drivers on dengue dynamics

We found strong evidence that temperature drives the spatial limits and temporal dynamics of dengue across Vietnam. The nonlinear effect of monthly $T_{mean}$, peaking around 27 °C, is consistent with

evidence from vector biology[26] and modelling studies[37]. Notably, temperature of the coolest month of the year explained the gradient in transmission intensity across Vietnam (Supp. Fig. 7), strongly suggesting that constraints on viral and mosquito persistence in cooler months are barriers to endemic establishment in the north. Indeed, phylogeographic studies have shown that transmission chains rarely persist over winter, and that yearly case surges in northern Vietnam (including Ha Noi) are mainly seeded by reintroductions from the south[42]. Consistent with this, we found markedly larger effect sizes and predictive influence of human mobility in the north, which may reflect the importance of higher connectivity in facilitating annual DENV reintroductions. Smaller effects of mobility in the south might reflect that populations are sufficiently large, and the climate consistently suitable, to support sustained local transmission. More generally, these results suggest that highly connected localities in climatically marginal regions may be useful targets for early surveillance, as dengue expansion is likely to proceed via establishment in these areas before radiating outward[65,66]. Our analyses were constrained by imprecise mobility metrics (Table 1), and more detailed data sources (such as transport networks or mobile phone data) could provide further insights into these expansion dynamics[15].

There is a need to understand how climatic variability and extreme weather events interact with local socioeconomic contexts to drive outbreak dynamics, rather than considering climate hazards as independent drivers[38,67]. Our finding that hydrometeorological dynamics are significant, multi-scalar drivers of dengue risk in Vietnam (Fig. 3) adds to an emerging evidence consensus for general effects of drought on dengue, as similar long-lag drought effects have also been observed in Latin America and the Caribbean[36,37]. Expanding on those studies, we found that high improved water supply coverage changes the functional shape of the dengue-SPEI-6 relationship, buffering against risk during low amplitude dry periods and sharply increasing risk during severe drought (Fig. 6). This is strongly indicative of a mediating role of water storage practices. During slightly dry periods, piped or borehole-derived water supply may decrease the need to store water in containers, and/or increase the frequency of stored water replacement, both of which reduce vector production rate[68]. In contrast, improved supply during drought may increase the availability and propensity to store water in containers, whereas households with lower access might switch to alternative sources such as bought water (as suggested by past research in the Mekong Delta[59]). Our water infrastructure metric did not include service reliability or sociocultural perceptions of water quality and reliability, all of which impact water usage and storage norms[59,69], and the inference of extreme drought effects by definition relied upon a relatively small number of observations (Fig. 6). Nonetheless, including this interaction improved spatial and temporal predictive accuracy, particularly in the highest-burden southern provinces (Supp. Fig. 18). This has implications for spatial prioritisation of interventions (e.g., vector control) to localities where water storage is highest during dry or drought periods, as well as highlighting that developing accurate local-scale dengue forecasts will likely need to account for complex climate-socioeconomic interactions.

## The role of environmental change in driving long-term dengue emergence in Vietnam

Reported dengue burden has grown in many regions of Vietnam over the last two decades, including northward expansion into central regions and the Red River Delta[11] (Fig. 1b). Our findings strongly suggest that recent socio-environmental and climatic changes have substantially contributed to this emergence trend, although our approach does not attribute observed trends to changes in specific drivers, and it is feasible that changing patterns of clinical awareness or healthcare access also played a role (Methods, Supp. Fig. 19). In future, age-stratified incidence or seroprevalence surveys could provide

additional evidence to further disentangle the drivers of these trends; relatively few such studies have been published over the last decade from Vietnam. Recent evidence is consistent with the patterns we observed (suggesting slight declines in transmission in the south[70] and increases in central areas[40]), but inference of changing force of infection from age-stratified data is challenging[71], highlighting this as an important future research area to understand dengue emergence trajectories in Vietnam.

Notably, while most studies of dengue and climate change have focused on future scenario-based projection[34,72], we instead used historical reanalysis data which suggests that climate change in recent decades has already expanded and redistributed transmission risk, likely facilitating dengue's northward spread (Fig. 5 and Supp. Fig. 15). In the north, the combined effects of a lengthening transmission season and rapid rises in mobility (up to 14-fold since 1998) have probably contributed substantially to the emergence of dengue as an annual problem in Ha Noi and the Red River Delta. Evidence of reductions in risk during the hottest months, however, suggest that future climate change will have complex effects on spatiotemporal patterns of dengue burden. Our approach stops short of attributing these effects to anthropogenic climate forcing, instead comparing present-day risk patterns to a reference period preceding the recent global temperature uptick; applying a formal detection and attribution framework will be an important next step towards quantifying the anthropogenic fingerprint on dengue burden[73].

Recent changes in infrastructure may have had complex effects on the landscape of dengue, with transmission risk simultaneously increased via widespread expansion of sanitation systems and reduced via the growth of cities and improvements in water supply. Indeed, rather than a simple positive dengue–urbanisation relationship, localities most vulnerable to outbreaks are probably peri-urban and transitional landscapes with increasingly dense populations but relatively weak infrastructure and services. Our findings consequently support improvements in hygienic water supply infrastructure as a pillar of climate adaptation to increasing mosquito-borne arboviral risks, but also highlight potential limits to this adaptation. Climatic changes are stressing water security in much of Southeast Asia, including Vietnam which has recently experienced severe droughts and saltwater incursion, and regional drought risks are projected to increase in future[74,75]. Expanding access to improved water supply infrastructure may mitigate dengue risks during dry periods, but might be insufficient to reduce dengue risks following severe droughts without additional improvements to household water security. More broadly, our study shows the value of integrating explanatory (hypothesis-driven) and predictive methods to understand the interacting effects of climate and socioeconomic factors on emerging diseases.

## Methods

### Dengue surveillance data
Since 1998 Vietnam has maintained a dedicated national dengue passive surveillance system. Data on monthly dengue case counts from May 1998 to April 2021 at administrative level-2 (*"districts"*) were collected and collated at the Pasteur Institute Ho Chi Minh City (Southeast and Mekong River Delta provinces), Pasteur Institute Nha Trang (Central coastal provinces), Tay Nguyen Institute of Hygiene and Epidemiology (Central Highlands provinces) and National Institute of Hygiene and Epidemiology in Ha Noi (Northern provinces), with time series beginning between 1998 and 2001 depending on the region (Supp. Fig. 1). Case counts comprised clinically diagnosed (i.e., suspected) cases from the passive surveillance system. Diagnostic guidelines followed the standardised WHO guidelines for dengue diagnosis (using the 1997 WHO definitions prior to 2009, and using the revised 2009 classifications from 2009 onwards), and were applied nationwide throughout the study period. Following the Vietnam

national guidelines for dengue prevention and control, 3% of clinically diagnosed cases were laboratory-confirmed using viral isolation techniques for DENV serotype monitoring, and additionally between 5% and 7% of cases were confirmed using serological tests (IgM antibody capture enzyme-linked immunosorbent assay (MAC-ELISA)). District-level data on lab-confirmation rates were not available so were not included in our analyses. Other arboviral infections, particularly Zika and chikungunya, could be clinically misdiagnosed as dengue; however, reported case numbers and seroprevalence estimates have been low (e.g., only 265 reported Zika cases between 2016-19)[76,77], so this would be unlikely to substantially impact inference. Currently, there are 713 districts in Vietnam, although a substantial number of these were established through redrawing of administrative boundaries since 1998; to ensure geographical comparability throughout the study period, we combined dengue case counts for 46 districts to match their 1998 boundaries, creating a final dataset of 174,936 monthly case counts from 667 districts (Fig. 1a and Supp. Fig. 1). Case counts were assigned to a dengue transmission year (from April to May) for modelling.

### Socio-environmental and climatic covariates
We developed spatially- and temporally-explicit covariates to represent hypothesised drivers of dengue transmission and spread (Table 1). Covariates are visualised in Supp. Figs. 2 and 3, and data sources and processing are summarised below (for full description see Supp. Table 1 and Supp. Text 1). Raster data extraction and processing was conducted using 'sf', 'raster', 'exactextractr', 'dplyr' and 'magrittr' in R 4.0.3[78–80]. We accessed population data (total and density) from census-based data, urbanisation metrics (built-up land extent, and expansion rate in preceding 3- and 10-year windows) from satellite data[48], and infrastructure metrics (% households with access to hygienic toilet, and % piped or borehole-derived water) from the Vietnam Population and Housing Census (2009 and 2019, interpolated and projected to annual values). We accessed province-level annual road travel rates (km per inhabitant reported by the Vietnam General Statistics Office) as a measure of observed levels of population movement. In the absence of detailed mobility data such as mobile phone records, we used parameter-free gravity and radiation models to predict annual district-level relative connectivity (predicted mean population flux), based on population data and pairwise travel times between all pairs of districts[16,81].

Monthly temperature indicators (monthly means of daily mean, minimum and maximum temperature, i.e., $T_{mean}$, $T_{min}$ and $T_{max}$) were derived from ERA5-Land reanalysis data[49,82,83]. Since broad climatic suitability gradients could confound relationships with other variables, we also calculated three annual temperature indicators to represent more fundamental constraints on dengue persistence (annual $T_{mean}$, annual mean $T_{min}$ and $T_{mean}$ of the coolest month, all from the same dengue year). Monthly precipitation indicators were derived from bias-adjusted ERA5 data (WFDE5; Supp. Figs. 3–4). In addition to precipitation, we used the R package 'spei'[51] to estimate multi-scalar drought indicators (Standardised Precipitation Evapotranspiration Index; SPEI) from 40 year timeseries of monthly WFDE5 precipitation and ERA5-Land potential evapotranspiration in each district (reference period 1981–2020). SPEI incorporates effects of both precipitation and evapotranspiration on water availability[52,53], with values above and below 0 indicating, respectively, surface water excess or deficit relative to the long-term historical average in a given seasonal time window (for example, a 6-month SPEI for Jan–Jun 2018 would compare to Jan–Jun in all other years). SPEI values denote the relative magnitude of this deviation, from near-normal to moderate (absolute values from 0 to 1), from moderate to severe (absolute values 1 to 2), to extreme wet/dry conditions (absolute values > 2). We estimated monthly SPEI within 1-month, 6-month and 12-month windows to capture varying timescales of drought (SPEI-1 as short-timescale; SPEI-6 and SPEI-12 as long-

timescale). We considered both short- and long-timescale drought indicators in the same model as they reflect likely different causal influences on dengue transmission (Methods) and were not strongly correlated (Pearson coefficients between SPEI-1 and SPEI-6 = 0.52; SPEI-1 and SPEI-12 = 0.37). Climatic covariates were derived at lags of 0 to 6 months prior to the focal month to account for delayed effects (Supp. Text 1).

## Statistical model development

To infer relationships between covariates and dengue incidence we fitted spatiotemporal models in a Bayesian framework (integrated nested Laplace approximation, in INLA 21.7.10.1[84,85]). Monthly dengue case counts $Y_{i,t}$ ($n = 174,936$) were modelled as a negative binomial process to account for overdispersion:

$$Y_{i,t} \sim \mathrm{NegBinom}(\mu_{i,t}, n) \tag{1}$$

where $n$ is the size (overdispersion) parameter and $\mu_{i,t}$ is the expected mean number of cases for district $i$ during month $t$, modelled as a log link function of the following general linear predictor:

$$\log(\mu_{i,t}) = \alpha + P_{i,t} + \rho_{r(i),t} + u_{i,y(t)} + v_{i,y(t)} \tag{2}$$

Here, $\alpha$ is the intercept and $P_{i,t}$ is log population included as an offset. $\rho_{r(i),t}$ is a province-specific effect of calendar month to account for geographic variability in dengue seasonality (districts $i$ are nested within 63 provinces $r$), specified as a cyclic first-order random walk to capture dependency between successive months. To account for unexplained variation in spatiotemporal patterns of dengue across Vietnam (due to unmeasured factors such as population immunity), $u_{i,y(t)}$ and $v_{i,y(t)}$ are dengue year-specific (23 years, $y$) spatially structured (conditional autoregressive; $u$) and unstructured (i.i.d; $v$) district-level random effects, jointly specified as a Besag-York-Mollie model[55].

We fitted the above random effects-only model as a baseline (Supp. Fig. 5), and conducted model selection to develop a multivariable model including population, climate, urbanisation, infrastructure and mobility covariates (Table 1). We compared models using within-sample information criteria: Watanabe-Akaike Information Criterion (WAIC), Deviance Information Criterion (DIC) and cross-validated logarithmic score (log-score; calculated from the pointwise conditional predictive ordinate, an approximation of leave-one-out cross-validation). Comparing variation explained between different models using metrics such as pseudo-$R^2$ was not particularly informative, as the district-level random effects ($u_{i,y(t)} + v_{i,y(t)}$) are at the same annual resolution as most covariates (Table 1), and thus tend to compensate for excluded variables. We instead calculated measures of unexplained random effects variation (mean absolute error in district-level or seasonal effects), which indicates how much these effects attenuate towards zero when covariates are included. We first selected each covariate's best-fitting type (either $T_{mean}$, $T_{min}$ or $T_{max}$ for monthly temperature; either SPEI-6 or SPEI-12 for long-timescale drought; gravity or radiation; 3- or 10-year urban expansion), functional form (linear, logarithmic or nonlinear, the latter specified as a second-order random walk) and lag (climate variables only) by adding each individually to the baseline model and comparing WAIC (Supp. Fig. 6). Covariates considered for inclusion in a full multivariable model were: fixed effects of $T_{mean}$ coolest month, log population density, log gravity flux, log road traffic per inhabitant, built-up land, log 10-year urban expansion rate, and nonlinear effects of hygienic toilet access, piped water access, $T_{mean}$ 1-month lag, SPEI-1 1-month lag and SPEI-6 5-month lag.

Owing to the dataset's large size and the expectation of confounding relationships among covariates, it was both undesirable and computationally unfeasible to conduct a programmatic covariate selection process. Instead, we conducted a more limited model comparison procedure to develop a final multivariable model. To do this, we first excluded covariates with evidence of substantial multicollinearity when tested using variance inflation factors (log population density and log gravity flux were highly collinear, and the former was excluded because gravity flux improved models more during individual covariate analysis; Supp Fig. 6). We then fitted a multivariable model including all 10 remaining covariates and compared this to 10 separate models each holding out 1 covariate at a time. Covariates whose inclusion did not improve model fit according to at least 2 of the 3 within-sample metrics (WAIC, DIC and log-score) were excluded. All covariates improved the model by this majority rule criterion, and were retained.

We examined residuals and conducted posterior predictive checks to check the model met distributional assumptions. We also conducted a sensitivity analysis based on degree of urbanisation because, despite relatively low collinearity overall (Supp. Fig. 2), many large cities cluster with relatively high values for many key covariates. Since this could affect parameter estimates, we tested sensitivity by sequentially re-fitting the model holding out all observations in areas with >90%, >70% or >50% of population residing in urban areas as defined from census data (i.e., fitting the model to data from increasingly rural settings). To examine whether socio-environmental effects differ substantially between endemic and emerging dengue transmission settings, we also separately fitted the final multivariable model to data from southern Vietnam (Mekong River Delta, Southeast, South Central Coast and Central Highlands) and northern Vietnam (North Central, Red River Delta, Northeast and Northwest). Although patterns of population mixing may have substantially differed in 2020 due to COVID-19-associated movement restrictions, any impacts on dengue incidence should be accounted for via the district-level random effects and traffic covariates (Fig. 2).

## Measuring covariate predictive influence through block cross-validation tests

Inference can be strengthened through combining explanatory and predictive approaches, for example by using structured predictive tests to challenge the ability of hypothesis-led explanatory models to predict unseen observations[56,57] (i.e., testing the generalisability of inferred relationships). For strongly spatially dependent phenomena such as disease incidence, block cross-validation designs—which holdout data in spatially- or temporally structured blocks—are more appropriate than fully randomised approaches[86], and can provide insights into how different variables contribute to predicting different dimensions of a phenomenon (Supp. Fig. 8). To estimate the influence of individual covariates on predicting spatial and temporal variability in dengue incidence, we conducted block cross-validation experiments to estimate out-of-sample (OOS) prediction error for the baseline model, full model, and 10 models each excluding a single covariate from the full model. In each run, the dataset was 5-fold partitioned (observations were randomly allocated to folds following a given block holdout design, as described below) and OOS predictions were generated for each model using 80%-20% train-test splits (i.e., across 5 submodels). Prediction error (difference between observed and predicted cases) was summarised as mean absolute error (MAE$_{OOS}$), across all observations, at district-level and, to examine differences between regions, across all observations within either southern or northern Vietnam (Supp. Figs. 11 and 12)

This procedure was repeated 10 times each for 3 block holdout designs[86] to account for variation associated with random allocation of folds (Supp. Fig. 8a, b). *Spatial*: 5-fold of complete districts, i.e., predicting full dengue incidence time series in completely unobserved areas. *Spatiotemporal:* 5-fold of district-year combinations, i.e., predicting completely unobserved years in partially observed locations. *Seasonal*: 5-fold of quarterly (3-month) blocks per-district, i.e.,

predicting unobserved intra-annual epidemic dynamics. Under spatial and spatiotemporal holdout designs, the expected magnitude of dengue incidence in unobserved locations and years is inferred from nearby observed locations, via the spatially structured effects $u_{i,y(t)}$ (Supp. Fig. 8c). These designs therefore test the contribution of covariates to predicting spatial heterogeneity in dengue incidence dynamics among nearby locations (i.e., differences from the expected similarity to neighbouring districts). Under the seasonal block design, the random effects contain information about the expected magnitude of cases in unobserved blocks, inferred from other observations in the same district and year (Supp. Fig. 8d). This design therefore tests the contribution of monthly climatic variability to predicting departures from this seasonal expectation (Supp. Fig. 8d).

### Examining the recent impacts of climate change using historical temperature data

To examine the possible contribution of recent climate change to dengue expansion patterns nationally, we used the inferred risk function of monthly temperature ($T_{mean}$ 1-month lag; Fig. 3e) to project monthly posterior mean temperature-driven dengue risk since 1950 for all districts, using long-term $T_{mean}$ data from ERA5-Land reanalysis. To do this, we used the fitted risk function to predict the monthly marginal effect of $T_{mean}$ on dengue incidence (i.e., while holding all other variables constant) for each month across the full historical time series. To visualise present-day risk dynamics in space and time, we then summarised and mapped 20-year means of monthly district-level risk for the period 2001–2020. To test for effects of climate change, we compared 20-year average risk between a historical reference period (1951–1970) and the present-day period, per-district and month, using linear models with time period as a categorical covariate. The use of 20-year averages was to account for natural climate system variability. The historical reference period was based on the earliest available ERA5-Land data, and while not reflective of the pre-industrial baseline, precedes the sharp acceleration of global temperatures that has occurred since around 1970. Climate reanalysis is based on assimilating observational data with climate models to provide a detailed and accurate reconstruction of historical climate dynamics, and its accuracy relies upon observational data. The number of observations assimilated by ERA5 increases tenfold between 1950 and 1970, and this lower data coverage might reduce accuracy in earlier years[58]; we therefore also tested the sensitivity of results to defining a later reference period (1971–1990) when coverage is much higher.

### Examining multi-scalar effects of drought, and interactions with infrastructure, in southern Vietnam

We extended the full model to investigate the effects of interactions between extreme wetness/drought and local infrastructure on dengue incidence, over multiple timescales and delays (from 0 to 6 months), focusing on endemic southern Vietnam. We examined the relationship between SPEI and dengue incidence, and improvements in model fit, for all lags and timescales of SPEI (SPEI-1 and SPEI-6 at 0 to 6 months delay), by including each metric individually in the full model containing all covariates except SPEI. To test the hypothesis that the effects of drought on dengue are mediated by water supply, we tested whether models were improved by stratifying the best-fitting short-timescale (SPEI-1 1-month) and long-timescale (SPEI-6 5-month) drought indicator by either level of improved water access or urbanisation (grouped as low, intermediate, or high, defined as <25%, 25–75% or >75%, respectively). We expected that stratification of the SPEI-6 effect by water supply would improve models more than stratifying SPEI-1, and that stratifying by water supply would improve models more than stratifying by urbanisation (Results). We used information criteria as described above (WAIC, DIC and log-score) to evaluate whether including each interaction improved model fit compared to the full (non-interaction) model. For each model, we also tested whether interactions reduced OOS prediction error under spatio-temporal and seasonal cross-validation, as described above.

### Reporting summary

Further information on research design is available in the Nature Portfolio Reporting Summary linked to this article.

## Data availability

Dengue surveillance data for a subset of 4 Vietnamese provinces (Dak Lak, Khanh Hoa, Ha Noi and Dong Nai) are provided in the study repository to demonstrate analysis pipeline functionality (https://doi.org/10.5281/zenodo.10159288). The nationwide dengue incidence data underlying these results are available from Phan Trong Lan, General Department of Preventive Medicine, MOH (phantronglan@gmail.com). All other data used in analyses were accessed from open sources. Land cover data was accessed from ESA-CCI (https://www.esa-landcover-cci.org/), census-based population, sanitation, housing and mobility data from the Vietnam General Statistics Office (https://www.gso.gov.vn/en/homepage/) and climate reanalysis data from Copernicus (ERA5-Land https://cds.climate.copernicus.eu/cdsapp#!/dataset/reanalysis-era5-land; WFDE5 https://cds.climate.copernicus.eu/cdsapp#!/dataset/derived-near-surface-meteorological-variables). Processed versions of these datasets used in analyses are provided in the study repository.

## Code availability

All data processing and modelling code used for this study are available at the study repository (https://doi.org/10.5281/zenodo.10159288).

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

## Acknowledgements

The authors thank Colin Carlson for the insightful discussion of early results. This research was supported by the UK Space Agency-funded International Partnership Programme 2 D-MOSS project *"An integrated dengue early warning system driven by Earth Observations in Vietnam"* (all authors), as well as a Royal Society Dorothy Hodgkin Fellowship (R.L.), a Wellcome Trust Sir Henry Wellcome Fellowship (206471/Z/17/Z) and a UK Medical Research Council Career Development Award (MR/V031112/1, both O.B.).

## Author contributions

Initial concept: R.G., O.B., R.L. Analysis design: R.G., F.C.-G., O.B., R.L. Data collection: P.T.L., P.T.H., V.S.N., V.T.D., D.T.H., N.T.D., V.C.C., L.T., D.K.Q., T.M.H., N.H.T., T.T.H. Data collation and processing: R.G,. V.T.D., N.T.D., L.T., D.K.Q., G.T., B.H. Data analysis and modelling: R.G. Writing (initial draft): R.G. Writing (review and editing): R.G., F.C.G., P.T.L., P.T.H., V.S.N., V.T.D., D.T.H., N.T.D., V.C.C., L.T.T.T., D.K.Q., T.M.H., N.H.T., T.T.H., G.T., E.A., Q.H., B.H., D.L., O.J.B., R.L. Funding: R.L., O.B., G.T., D.L. Project administration: G.T., D.L.

## Competing interests

The authors declare no competing interests.

## Additional information

[1]Department of Infectious Disease Epidemiology & Dynamics, Faculty of Epidemiology and Population Health, London School of Hygiene & Tropical Medicine,
London, UK. [2]Centre for Mathematical Modelling of Infectious Diseases, London School of Hygiene and Tropical Medicine, London, UK. [3]Centre on Climate
Change and Planetary Health, London School of Hygiene and Tropical Medicine, London, UK. [4]Centre for Biodiversity and Environment Research, Department
of Genetics, Evolution & Environment, University College London, London, UK. [5]Data for Science and Health, Wellcome Trust, London, UK. [6]General
Department of Preventative Medicine (GDPM), Ministry of Health, Hanoi, Vietnam. [7]National Institute of Hygiene and Epidemiology (NIHE), Hanoi, Vietnam.
[8]Pasteur Institute Nha Trang, Nha Trang, Khanh Hoa Province, Vietnam. [9]Tay Nguyen Institute of Hygiene and Epidemiology (TIHE), Buon Ma Thuot, Dak Lak
Province, Vietnam. [10]Pasteur Institute Ho Chi Minh City, Ho Chi Minh City, Vietnam. [11]Center for Disease Control, Dong Nai Province, Vietnam. [12]Center for
Disease Control, Khanh Hoa Province, Vietnam. [13]Center for Disease Control, Hanoi, Vietnam. [14]HR Wallingford, Wallingford, Oxfordshire, UK. [15]Barcelona
Supercomputing Center (BSC), Barcelona, Spain. [16]Catalan Institution for Research and Advanced Studies (ICREA), Barcelona, Spain. [17]These authors
contributed equally: Oliver J. Brady, Rachel Lowe. ✉e-mail: rory.gibb.14@ucl.ac.uk

