## [Peer Review File · Nature Communications]

Interactions between climate change, urban infrastructure and mobility are driving dengue emergence in VietnamREVIEWER COMMENTS

Reviewer #1 (Remarks to the Author):

This study investigates the change in the span of dengue throughout Vietnam over a 22-year period based primarily on factors relating to the environment and human development. The authors used Bayesian models and identified expanded transmission correlating with increased temperatures. The relationship between urban expansion and dengue emergence was found to be complex, identifying aspects of urban development that are associated with higher and lower dengue risk. The analyses presented are thorough and of overall high quality.

Specific comments are below.

1. Lines 159-160: This sample size differs from the total of having 22 years among 667 districts. Are the missing values months with zero incidence, months with no report, or months when some of the 667 districts did not yet exist?

2. Lines 198-199: The authors mention that mobility declined in 2020 due to COVID-19 restrictions. Are there any concerns that this last year of observation (May 2020 – Apr 2021) is affecting the model results due to substantially different circumstances? Restrictions to prevent COVID-19 led to less mobility, potential halts in urban development, and lower population mixing. It has been noted that dengue cases also decreased substantially. It would be beneficial to discuss whether this year was an outlier that affects model results or if the model is robust to drastically changing circumstances.

3. Is the fitted model a multivariate model or a multivariable model? Multivariate models refer to those with a multivariate outcome, with the dependent variable being a vector rather than a single value. Hidalgo and Goodman (2013, AJPH) describe this in detail.

4. Lines 226-227 The full fitted model includes both SPEI in a 1-month comparison (with a 1-month lag) and in a 6-month comparison (with a 5-month lag). My understanding of this is that the model uses, for month m , the comparison in SPEI spanning months $m-1$ through $m-2$ as well as the comparison spanning months $m-5$ through $m-11$. The desire to consider multiple spans and lags has clear motivation, but there may be potential concerns that including more than one contributes highly redundant information. The authors should justify inclusion of more than one SPEI predictor.

5. 398-401: This is a very interesting finding and a good point that, despite a presumed reduction in standing water for mosquito habitats/breeding, flush toilet access was associated with higher dengue risk. Is it possible that access to flush toilets, instead of primarily representing a reduction in standing water, serves as a proxy for other factors that could contribute to dengue risk or case detection? One example could be that flush toilet access is an indicator for a lack of poverty, and therefore greater ability to seek healthcare when sick and detect dengue infections.

6. Lines 514-516: Is there information regarding how many cases were confirmed/suspected? Are there differences in reporting confirmed/suspected cases over time?

7. Additional detail regarding temperature data is needed. What is the temporal resolution of T_{mean} , T_{min} , and T_{max} that are being averaged each month? Does the annual mean of the coolest month refer to calendar years or May-Apr transmission years?

8. The first line of Table 1 describes Annual temperature. Should the leftmost cell read "mean of coolest month?"

Reviewer #2 (Remarks to the Author):

Vietnam have observed recent changes in dengue epidemiology which raises the need for investigating factors that influences its transmission. The study presented here attempts to do just that, identifying potential effects on dengue risks related to climate, urban infrastructure, hydrometeorology changes. Bayesian spatiotemporal regression models and subsequent analysis used here are adequate and reasonable. I find the results presented in this study are fascinating and practical. The finding of peak incidence in transitional landscapes is interesting and have been observed for other mosquito-borne viruses (10.1038/s41559-017-0108). Decrease of dengue risk in piped access water region has great implications on policy. Other findings confirm the importance of human movement and climate on the transmission of the disease. The manuscript here was well written and structured, and in my opinion, is ready for publications with minimal changes.

I am aware that the authors have considered potential bias of the data collection in lines 171-174. However, it will be good to further discuss the limitations of the dataset elsewhere in the manuscript. Are there any changes in the reporting of cases over time or between the districts? Are there any changes in the cases definition over time?

Figure 2: The population density would be clearer if the plotted in log10 scale.

Reviewer #3 (Remarks to the Author):

This is a well-written manuscript that elegantly outlines the complexities of relating climatological factors to dengue risk, given regional and cultural differences in water storage, mobility, urban development, and access to water / sanitation infrastructure. The rationale and findings are well-reasoned and result in impactful findings; among them:

- Tmean of the coolest month explained 50% of the variation in district-level random effects (indicating that, as nicely shown in Figure 5b, temperature may be an important predictor of the observed increases in dengue transmission in recent decades).
- Spatial predictors differed between the north and the south, with mobility being a top predictors in the dengue-emergent north (indicating that seeding of new dengue strains may be important for maintaining the burden of dengue in some regions, but not all).
- Drought and water infrastructure had complex relationships with dengue risk, likely related to differential water storage behaviors in different severities of drought.

Major comments:

- The introduction is quite complex but, to this reviewer, appropriately so – these relationships and questions are complex (and important to disentangle)
- It seems a strong assumption that ‘the pronounced geographical pattern and absence of obvious step function suggest that these trends are unlikely to be solely driven by specific changes in surveillance or diagnostic practices...’ Is increasing availability of diagnostics, or increased clinical suspicion for dengue in northern provinces (for example) necessarily a step function? What if NS1 testing, for example, were gradually expanded across the north in recent years – could this be consistent with the data as well? This bears a bit more exploration and explanation. Relatedly, if the data are available on the proportion of cases that were diagnosed clinically (suspected) versus lab-confirmed cases by year, that may be helpful.

Minor comments:

- Please briefly define more technical or abstract terms such as ‘gravity or radiation flux’ and what is meant by ‘hygienic toilet’, for example. This would be helpful for a diverse readership.
- Was elevation considered in the analysis? (Is that part of the land use analyses?) If not, please comment on possible implications of elevation in spatial patterns of risk of dengue in Vietnam.
- Are there any information on the prevalence of air-conditioning units? This reviewer wonders whether that may relate to decreased risk observed with very high temps (at least in more developed places like Ho Chi Minh City...)

- Please report the estimated numbers of Zika cases identified in 2015-17 – certainly reported Zika cases are likely low now, but may have been higher during the pandemic. If significant numbers of Zika cases, please comment on possible impacts on your analysis.
- An additional (complementary) piece of data to support increasing force of infection, if available, would be age-stratified incidence data by region and by decade.

Response to reviewer comments for Gibb et al, “Interactions between climate change, urban infrastructure and mobility are driving dengue emergence in Vietnam”
Nature Communications, Oct 2023

We thank the reviewers for their overall positive comments and the helpful suggestions to improve the manuscript’s clarity and rigour. We have addressed these comments in full in the point-by-point responses below, and we have pasted any substantial updates to the text after each response. Line numbers refer to the revised manuscript.

Reviewer #1 (Remarks to the Author):

This study investigates the change in the span of dengue throughout Vietnam over a 22-year period based primarily on factors relating to the environment and human development. The authors used Bayesian models and identified expanded transmission correlating with increased temperatures. The relationship between urban expansion and dengue emergence was found to be complex, identifying aspects of urban development that are associated with higher and lower dengue risk. The analyses presented are thorough and of overall high quality.

Thank you for the positive comments and helpful suggestions on the manuscript. Please see responses to specific comments below.

Specific comments are below.

1. Lines 159-160: This sample size differs from the total of having 22 years among 667 districts. Are the missing values months with zero incidence, months with no report, or months when some of the 667 districts did not yet exist?

The surveillance time series data started in different years depending on the region of the country, with the earliest start in the northern regions (1998) and districts in the southern regions commencing between 1999 and 2001, depending on the province. In each district, the period prior to the commencement of surveillance was not included in the dataset; this is why the total sample size differs from a full 22 years by 667 districts. This is visualised in Supp. Figure 1 and stated in lines 520-521, although we have added an extra note in the legend of Figure 1 (lines 755-757) for extra clarity around when the surveillance commenced.

2. Lines 198-199: The authors mention that mobility declined in 2020 due to COVID-19 restrictions. Are there any concerns that this last year of observation (May 2020 – Apr 2021) is affecting the model results due to substantially different circumstances? Restrictions to prevent COVID-19 led to less mobility, potential halts in urban development, and lower population mixing. It has been noted that dengue cases also decreased substantially. It would be beneficial to discuss whether this year was an outlier that affects model results or if the model is robust to drastically changing circumstances.

This is a good question, although we are not concerned about the potential for 2020 to substantially impact the results, for three reasons: (1) 2020 did not see a uniform drop in dengue cases throughout the country, with case incidence still relatively high – i.e. comparable to previous years – in the Southeast and Central regions (see Supp. Figure 1); (2) Any substantial variation between years driven by unmeasured factors (including for 2020) will be mainly accounted for by the spatiotemporal random effects structure within the model; and (3) An earlier version of the analyses was developed using data only up to 2019, and results were qualitatively the same in terms of inferred slope parameters and nonlinear effects. We agree it is important to clarify this, so we have

added a sentence in the Methods to highlight the robustness of the model to different patterns of mixing in 2020 (lines 634-637):

New text: "Although patterns of population mixing may have substantially differed in 2020 due to COVID-19 associated movement restrictions, any impacts on dengue incidence should be accounted for via the district-level random effects and traffic covariates (Figure 2)."

3. Is the fitted model a multivariate model or a multivariable model? Multivariate models refer to those with a multivariate outcome, with the dependent variable being a vector rather than a single value. Hidalgo and Goodman (2013, AJPH) describe this in detail.

This is an important point, thank you, and an error on our part. It is a multivariable model, and we have corrected the MS throughout to clarify this.

4. Lines 226-227 The full fitted model includes both SPEI in a 1-month comparison (with a 1-month lag) and in a 6-month comparison (with a 5-month lag). My understanding of this is that the model uses, for month m , the comparison in SPEI spanning months $m-1$ through $m-2$ as well as the comparison spanning months $m-5$ through $m-11$. The desire to consider multiple spans and lags has clear motivation, but there may be potential concerns that including more than one contributes highly redundant information. The authors should justify inclusion of more than one SPEI predictor. *We considered both short- and long-timescale SPEI metrics in the same model to reflect different potential causal influences on dengue transmission, one via immediate accumulation or drying of breeding sites (short-term) and one potentially mediated by human behaviour in response to drought (long-term). Importantly, due to the substantial differences in timescale these metrics (SPEI-1 and either SPEI-6 or SPEI-12) are not strongly correlated to each other – i.e. there is minimal information redundancy between them, as it is possible to have a 1-month period of transient rainfall excess (positive SPEI-1) during a long-term drought period (negative SPEI-6) and vice-versa. The rationale for including both was already stated in the Results section but we agree it would be helpful to clarify the lack of information redundancy, so we have added an extra sentence to the Methods to ensure this is well-explained (lines 572-574):*

New text: "We considered both short- and long-timescale drought indicators in the same model as they reflect likely different causal influences on dengue transmission (Methods) and were not strongly correlated (Pearson correlation coefficients between SPEI-1 and SPEI-6 = 0.52; SPEI-1 and SPEI-12 = 0.37)."

5. 398-401: This is a very interesting finding and a good point that, despite a presumed reduction in standing water for mosquito habitats/breeding, flush toilet access was associated with higher dengue risk. Is it possible that access to flush toilets, instead of primarily representing a reduction in standing water, serves as a proxy for other factors that could contribute to dengue risk or case detection? One example could be that flush toilet access is an indicator for a lack of poverty, and therefore greater ability to seek healthcare when sick and detect dengue infections.

This is a good point and something we had not considered in as much detail. It is indeed possible that improved access to flush toilets is in part proxying for economic improvements (and consequent health care access changes) that improve detection capacities for dengue, and this might explain why it provides such a substantial improvement in the models. We have added a clause to the Discussion to highlight this as a possibility (lines 400-401).

Updated text: "It is possible that this metric may instead index "urban-like" water-related infrastructure that provide amenable Aedes breeding habitat, such as storm drains, septic and water storage tanks

(although this relationship was robust in a model fitted only to data from rural districts; Supp. Figure 10), and/or proxy for economic improvements and associated increases in healthcare access that could increase case detection”.

6. Lines 514-516: Is there information regarding how many cases were confirmed/suspected? Are there differences in reporting confirmed/suspected cases over time?

Under the Vietnam national guidelines for dengue prevention and control, 3% of clinically diagnosed cases were confirmed via virus isolation techniques to monitor the circulation of different DENV serotypes, while between 5% and 7% of cases were confirmed using serological tests (MAC-ELISA). These guidelines were implemented annually and nationwide from 1999 onwards so, importantly, the reporting protocols were consistent throughout almost the entire study period. The Vietnam dengue surveillance data system does not keep data on the proportion of lab-confirmed versus clinically diagnosed cases at the same granularity (i.e. district-by-month) as the clinical surveillance data, so we are not able to provide a more detailed breakdown in space and time. We agree it is important to highlight the strengths and limitations of the data, so we have added text to the Methods and Results to highlight these points (lines 168-175 and 521-530).

Updated Methods text: *“Standardised WHO dengue diagnosis guidelines and lab-confirmation practices were applied nationally throughout the study period, with no obvious step change in the data in 2009 when the WHO definitions were changed (Methods), suggesting that these trends are unlikely to be driven by a specific change in surveillance or diagnostic practices. The trends may still in part be driven by reporting factors, such as increased clinical suspicion or healthcare access, although the pronounced geographical pattern and visual indications of a shift towards endemic dynamics in southern central regions (Supp. Figure 1) are strongly suggestive of true expansion.”*

Updated Results text: *“Case counts comprised clinically-diagnosed (i.e. suspected) cases from the passive surveillance system. Diagnostic guidelines followed the standardised WHO guidelines for dengue diagnosis (using the 1997 WHO definitions prior to 2009, and using the revised 2009 classifications from 2009 onwards), and were applied nationwide throughout the study period. Following the Vietnam national guidelines for dengue prevention and control, 3% of clinically-diagnosed cases were laboratory-confirmed using viral isolation techniques for DENV serotype monitoring, and additionally between 5% and 7% of cases were confirmed using serological tests (MAC-ELISA). District-level data on lab confirmation rates were not available so were not included in our analyses.”*

7. Additional detail regarding temperature data is needed. What is the temporal resolution of T_{mean}, T_{min}, and T_{max} that are being averaged each month? Does the annual mean of the coolest month refer to calendar years or May-Apr transmission years?

T_{mean}, T_{min}, and T_{max} are respectively the averages of daily mean, minimum and maximum temperatures respectively, calculated originally from processing hourly ERA5-Land temperature timeseries. A full description is in Supplementary Text 1, but we now have ensured this is clearer in the main text (lines 554-555). Annual mean temperature metrics are for the same “dengue year” (May-Apr), and we have again made line edits to clarify this in the text (lines 558-559).

8. The first line of Table 1 describes Annual temperature. Should the leftmost cell read “mean of coolest month?”

We considered three annual temperature metrics: mean annual temperature (T_{mean}), mean of monthly T_{min}, and T_{mean} of the coolest month, and the latter was selected for the model. We have now included all three in the leftmost cell of the table to ensure clarity for a reader (Table 1).

Reviewer #2 (Remarks to the Author):

Vietnam have observed recent changes in dengue epidemiology which raises the need for investigating factors that influences its transmission. The study presented here attempts to do just that, identifying potential effects on dengue risks related to climate, urban infrastructure, hydrometeorology changes. Bayesian spatiotemporal regression models and subsequent analysis used here are adequate and reasonable. I find the results presented in this study are fascinating and practical. The finding of peak incidence in transitional landscapes is interesting and have been observed for other mosquito-borne viruses (10.1038/s41559-017-0108). Decrease of dengue risk in piped access water region has great implications on policy. Other findings confirm the importance of human movement and climate on the transmission of the disease. The manuscript here was well written and structured, and in my opinion, is ready for publications with minimal changes.

Many thanks for the positive comments and we are glad you appreciated the manuscript.

I am aware that the authors have considered potential bias of the data collection in lines 171-174. However, it will be good to further discuss the limitations of the dataset elsewhere in the manuscript. Are there any changes in the reporting of cases over time or between the districts? Are there any changes in the cases definition over time?

We agree it is important to provide a breakdown of the diagnostic and reporting timelines to better understand how the data were generated and their limitations. The Vietnam national guidelines for dengue diagnosis and management were based on the WHO guidelines: until 2009 the case definitions for diagnosis followed the 1997 WHO guidelines (dengue fever and dengue grades 1-IV), and from 2009 onwards they followed the revised 2009 WHO guidelines (dengue, dengue with warning signs and severe dengue). These were applied nationally for the study period so, while it is possible that regional differences in reporting patterns existed due to e.g., differences in clinical awareness, standardised definitions and diagnostic guidance were in place throughout the timeline of our study. We have added additional text to clarify this for a reader (lines 168-175 and 521-530).

Updated Methods text: *“Standardised WHO dengue diagnosis guidelines and lab-confirmation practices were applied nationally throughout the study period, with no obvious step change in the data in 2009 when the WHO definitions were changed (Methods), suggesting that these trends are unlikely to be driven by a specific change in surveillance or diagnostic practices. The trends may still in part be driven by reporting factors, such as increased clinical suspicion or healthcare access, although the pronounced geographical pattern and visual indications of a shift towards endemic dynamics in southern central regions (Supp. Figure 1) are strongly suggestive of true expansion.”*

Updated Results text: *“Case counts comprised clinically-diagnosed (i.e. suspected) cases from the passive surveillance system. Diagnostic guidelines followed the standardised WHO guidelines for dengue diagnosis (using the 1997 WHO definitions prior to 2009, and using the revised 2009 classifications from 2009 onwards), and were applied nationwide throughout the study period. Following the Vietnam national guidelines for dengue prevention and control, 3% of clinically-diagnosed cases were laboratory-confirmed using viral isolation techniques for DENV serotype monitoring, and additionally between 5% and 7% of cases were confirmed using serological tests (MAC-ELISA). District-level data on lab confirmation rates were not available so were not included in our analyses.”*

Figure 2: The population density would be clearer if the plotted in log10 scale.

We respectfully disagree, having originally tried visualising population density on the log scale (as

this would have been consistent with how this covariate was transformed for modelling). Transforming to the logarithmic scale improves the ability to distinguish between each line for regions with lower population density, but at the expense of much less clearly showing the population growth trends in the higher-population regions (as the trends are visually flattened). Since the goal of this figure is to show the extent of socio-environmental change trends that have occurred during the study period, we feel that visualising on the identity scale is more visually intuitive and shows the trend more clearly.

Reviewer #3 (Remarks to the Author):

This is a well-written manuscript that elegantly outlines the complexities of relating climatological factors to dengue risk, given regional and cultural differences in water storage, mobility, urban development, and access to water / sanitation infrastructure. The rationale and findings are well-reasoned and result in impactful findings; among them:

- Tmean of the coolest month explained 50% of the variation in district-level random effects (indicating that, as nicely shown in Figure 5b, temperature may be an important predictor of the observed increases in dengue transmission in recent decades).
- Spatial predictors differed between the north and the south, with mobility being a top predictors in the dengue-emergent north (indicating that seeding of new dengue strains may be important for maintaining the burden of dengue in some regions, but not all).
- Drought and water infrastructure had complex relationships with dengue risk, likely related to differential water storage behaviors in different severities of drought.

Thank you for the positive comments and we are glad you agree the findings are interesting and impactful.

Major comments:

- The introduction is quite complex but, to this reviewer, appropriately so – these relationships and questions are complex (and important to disentangle)

To clarify the Introduction for a general readership we have made some minor structural changes to ensure a clear narrative (lines 76-88).

- It seems a strong assumption that ‘the pronounced geographical pattern and absence of obvious step function suggest that these trends are unlikely to be solely driven by specific changes in surveillance or diagnostic practices...’ Is increasing availability of diagnostics, or increased clinical suspicion for dengue in northern provinces (for example) necessarily a step function? What if NS1 testing, for example, were gradually expanded across the north in recent years – could this be consistent with the data as well? This bears a bit more exploration and explanation. Relatedly, if the data are available on the proportion of cases that were diagnosed clinically (suspected) versus lab-confirmed cases by year, that may be helpful.

We agree that a better understanding of how changing surveillance and diagnostic procedures shapes macro-scale patterns of apparent dengue emergence is important, and that any such changes would not necessarily manifest as a step change in the incidence data. We have therefore included some extra discussion of these changes in the manuscript (lines 168-175 and 521-530). In particular, we have emphasised that the same standardised guidelines for dengue diagnosis, management and lab testing were in place nationally throughout the study period. As we highlighted in our response to other reviewers, national guidance for clinical dengue diagnosis followed the 1997 WHO definitions before 2009, and the 2009 WHO guidelines from 2009 onward, and these were applied nationwide. The protocols for lab-confirmation were also applied nationwide from

1999 onwards: annually, 3% of clinically-diagnosed cases were confirmed via viral isolation for DENV serotype monitoring, while between 5% and 7% of cases were serologically confirmed using MAC-ELISA. As the data on the number (proportion) of lab-confirmed cases are not recorded within the dengue surveillance system data, we are unable to provide a breakdown by year.

It is certainly feasible that the expanding pattern of dengue cases is, at least partially, driven by increasing clinical suspicion and/or health system access, and we have ensured this is clearer in the text (lines 474-484). However, the epidemiological pattern of dengue outbreak dynamics over time does, in many places, appear consistent with true emergence rather than solely a detection trend. For example, in the central coastal and highland regions (i.e., Ha Noi and surrounding areas), there is some visual indication of a shift from sporadic isolated outbreaks to the kinds of multi-year epidemic cycles that are seen in endemic regions (Supp Figure 1). Rather than suggesting solely either a step change or gradual increase in diagnosis, these are suggestive of a true change in transmission dynamics. We have included some additional text to highlight this (lines 168-175).

Importantly, from the perspective of the inference of drivers we are undertaking in this study, such changes in detection patterns are unlikely to substantially impact the inference of climatic and socio-environmental relationships, as they will largely be captured by the spatiotemporal random effects (which account for unexplained interannual variability). To highlight this point, we have included an additional supplementary figure showing the yearly random effects, to show remaining unexplained variability, some of which will be caused by detection processes (Supp Figure 19).

Updated Results text (lines 168-175): *“Standardised WHO dengue diagnosis guidelines and lab-confirmation practices were applied nationally throughout the study period, with no obvious step change in the data in 2009 when the WHO definitions were changed (Methods), suggesting that these trends are unlikely to be driven by a specific change in surveillance or diagnostic practices. The trends may still in part be driven by reporting factors, such as increased clinical suspicion or healthcare access, although the pronounced geographical pattern and visual indications of a shift towards endemic dynamics in southern central regions (Supp. Figure 1) are strongly suggestive of true expansion.”*

Updated Methods text (lines 521-530): *“Case counts comprised clinically-diagnosed (i.e. suspected) cases from the passive surveillance system. Diagnostic guidelines followed the standardised WHO guidelines for dengue diagnosis (using the 1997 WHO definitions prior to 2009, and using the revised 2009 classifications from 2009 onwards), and were applied nationwide throughout the study period. Following the Vietnam national guidelines for dengue prevention and control, 3% of clinically-diagnosed cases were laboratory-confirmed using viral isolation techniques for DENV serotype monitoring, and additionally between 5% and 7% of cases were confirmed using serological tests (MAC-ELISA). District-level data on lab confirmation rates were not available so were not included in our analyses.”*

Updated Discussion text (lines 474-478): *“Our findings strongly suggest that recent socio-environmental and climatic changes have substantially contributed to this emergence trend, although our approach does not attribute observed trends to changes in specific drivers, and it is also feasible that changing patterns of clinical awareness or healthcare access also played a role (Methods, Supp. Figure 19).”*

Minor comments:

- Please briefly define more technical or abstract terms such as ‘gravity or radiation flux’ and what is meant by ‘hygienic toilet’, for example. This would be helpful for a diverse readership.

This is a great suggestion, thank you. We have now ensured that Table 1 provides full definitions of the covariates and is referred to properly in the text, and have defined specifically several terms in the main text.

- Was elevation considered in the analysis? (Is that part of the land use analyses?) If not, please comment on possible implications of elevation in spatial patterns of risk of dengue in Vietnam. In the early stages of the study we did consider elevation, and accessed district-level elevation data, but did not use it in the study. This is mainly because elevation is, locally within regions, strongly correlated to both air temperature (decreasing with altitude) and mobility (i.e. generally lower accessibility of more remote, high altitude regions). Since air temperature and accessibility are the two main causal pathways through which we expected elevation to impact dengue transmission, we elected to focus on those drivers directly.

- Are there any information on the prevalence of air-conditioning units? This reviewer wonders whether that may relate to decreased risk observed with very high temps (at least in more developed places like Ho Chi Minh City...)

This is a good question and it does appear that, nationally, the demand for air conditioning units has steadily grown over the last decade (<https://www.statista.com/statistics/909711/vietnam-ac-demand-units/>). It is feasible that behavioural factors these might in part lead to declining risk during extremely hot periods, although given highly variable levels of access to technology like air conditioning, we feel that the thermal biology of DENV within *Aedes* mosquitoes offers a more parsimonious biological explanation for the sharp decrease in transmission above T_{mean} of $\sim 27^{\circ}\text{C}$. Combined empirical and theoretical work has shown that temperature increases above a thermal optimum lead to decreased net transmission potential through, in particular, increasing the rate of mosquito mortality (<https://onlinelibrary.wiley.com/doi/pdf/10.1111/ele.13335>); such population-level impacts are likely to be the main driver of the strong nonlinear relationship we observe from the data.

- Please report the estimated numbers of Zika cases identified in 2015-17 – certainly reported Zika cases are likely low now, but may have been higher during the pandemic. If significant numbers of Zika cases, please comment on possible impacts on your analysis.

The number of Zika cases reported between 2016-19 was 265, predominantly in southern Vietnam (<https://pubmed.ncbi.nlm.nih.gov/32393198/>). Even with the possibility of a proportion of misdiagnosis or non-reported cases, this suggests that the Zika epidemic in Vietnam was fairly small, especially in comparison to the very large number of dengue cases reported during the same period. This suggests that any impacts on our dengue analysis, either via misdiagnosis or cross-immunity affecting epidemic dynamics, would be very small. We have specified this number in the Methods (line 532).

Updated Methods text (lines 529-533): *“Other arboviral infections, particularly Zika and chikungunya, could be clinically misdiagnosed as dengue; however, reported case numbers and seroprevalence estimates have been low (e.g. only 265 reported Zika cases between 2016-16)^{76,77}, so this would be unlikely to substantially impact inference.”*

- An additional (complementary) piece of data to support increasing force of infection, if available, would be age-stratified incidence data by region and by decade.

We agree that age-stratified incidence data would be valuable as an additional line of evidence to support our findings, although given the substantial challenges involved in understanding the drivers of changes in the age distribution of cases (e.g.

<https://www.pnas.org/doi/10.1073/pnas.2115790119>) we feel that doing such an analysis justice is unfortunately beyond the scope of this paper. However, we have included some extra discussion of this point (lines 478-484) as we feel it is important to highlight some of the published evidence that does exist. For example, age-stratified data for south Vietnam showed evidence for a declining trend in incidence rates between 2005 and 2015 and an upward trend in mean age of infection for clinical cases, which are suggestive of slight declines in FOI (consistent with our projections suggesting some evidence for slight declines in transmission in the south; Figure 5)

(<https://www.ncbi.nlm.nih.gov/pmc/articles/PMC9994699/>). Conversely, in Nha Trang (South Central where our study suggests that dengue is emerging) there is some evidence that the mean age of clinical cases has shown a slight declining trend since 2012, which is suggestive of increasing FOI (<https://www.ajtmh.org/view/journals/tpmd/98/2/article-p402.xml>). As we noted above there are substantial challenges to interpreting changes in age structure of incident cases. Notably, there are only a very small number of published age-stratified incidence or seroprevalence studies from Vietnam over the last decade, highlighting the value of this topic for follow-up study.

Updated Discussion text (lines 478-484): *“In future, age-stratified incidence or seroprevalence surveys could provide additional evidence to further disentangle the drivers of these trends; relatively few such studies have been published over the last decade from Vietnam. Recent evidence is consistent with the patterns we observed (suggesting slight declines in transmission in the south⁷⁰ and increases in central areas⁴⁰), but inference of changing force of infection from age-stratified data is challenging⁷¹, highlighting this as an important future research area to understand dengue emergence trajectories in Vietnam.”*

REVIEWERS' COMMENTS

Reviewer #1 (Remarks to the Author):

Thank you for responding to the previous comments. The authors have adequately addressed my previous comments, and the manuscript is suitable for publication in its present form.

Reviewer #3 (Remarks to the Author):

The authors have addressed all previous concerns from this reviewer, thank you